# Limited global effect of climate-Greenland ice sheet coupling in NorESM2 under a high-emission scenario

Konstanze Haubner[1], Heiko Goelzer[2], and Andreas Born[1]

[1]Department of Earth Science, University of Bergen, Bjerknes Centre for Climate Research, Bergen, Norway
[2]NORCE Norwegian Research Centre, Bjerknes Centre for Climate Research, Bergen, Norway

**Correspondence:** Konstanze Haubner (Konstanze.Haubner@uib.no)

**Abstract.** The Greenland ice sheet is melting at an accelerating rate due to the warming climate. In order to understand the potentially important ice-climate feedback processes, evolving ice sheets need to be included in global climate models. Here, we present results from the first bi-directional coupling of the Earth System model NorESM2 with the ice sheet model CISM2 for the Greenland ice sheet under an extended high emission SSP5-8.5 forcing from 1850 to 2300. In our simulation, the ice mass loss between 1850 and 2300 is equivalent to 1.4 m of sea-level rise. Comparing simulation results to an otherwise identical simulation with a fixed Greenland ice sheet, we see the same global trends in air, ocean and sea ice changes. The main signals are a $10°$ C global air temperature increase from 2000 to 2300, a reduced maximum AMOC at 26.5°N from average 23 to 9 Sv and an all-year free Arctic by 2200. Similar to other coupled CMIP models, the warming trend dominates the changes of the climate components. At the regional scale, elevation changes become an important part of the Greenland surface mass balance, accounting for 20% of the SMB change by 2200 and for 49% in 2300. By the year 2300, the ablation area covers 93% of the ice area. With a low climate sensitivity and relatively weak polar amplification in NorESM2, these results are on the lower end of the spectrum of expected ice mass loss under CMIP6 model forcing.

## Plain Language Summary

With warming air temperatures, the ice in Greenland is currently melting increasingly leading to a smaller extent and lowering of the ice surface. The melted ice runs as freshwater into the fjords and impacts ocean circulation. The lowering of the ice surface leads to higher temperatures and to more melt. It is still quite uncertain on which magnitude they are impacting each other. In this study, we add a new dynamic component - an ice sheet model simulating changes of the Greenland ice sheet - to an Earth system model that already captures the global climate evolution including ocean, atmosphere, land and sea ice. We find that under a strong warming scenario (SSP5-8.5), the air temperature warming of $10°$ C over 250 yrs is dominating the climate evolution and the impact that the added changes of the Greenland ice sheet bring is only affecting the local climate in Greenland. Yet, those changes and feedbacks are impacting the evolution of the Greenland ice sheet. Hence, ice-climate feedbacks should be considered when simulating Greenland ice sheet changes beyond 2100.

# 1 Introduction

The Greenland ice sheet's increasing mass loss (The IMBIE Team, 2020) is driven by oceanic (Straneo and Heimbach, 2013; Wood et al., 2021) and atmospheric warming (Bevis et al., 2019; Sellevold and Vizcaíno, 2020). Recent ice mass changes have been studied extensively (Khan et al., 2015; Machguth et al., 2016; McMillan et al., 2016; Mouginot et al., 2019; The IMBIE Team, 2020). However, Greenland ice sheet projections show a wide spread of mass evolution mainly due to climate forcing uncertainty (Goelzer et al., 2020; Holube et al., 2022), making Greenland a highly uncertain component for sea-level change projections by 2100 (Oppenheimer et al., 2019; Edwards et al., 2021). The flow of ice, depending on surface slope, conditions at the bed and ice rheology, delivers ice to the margins where it either melts or is calved as icebergs into the ocean. This mass loss is controlled by ocean and air induced melt and feedback mechanisms.

An important process at the ice-ocean interface are plumes of upwelling fresh-water runoff that lead to increased subsurface melt and undercutting at marine-terminating glaciers (Slater et al., 2017, 2021) which affects both calving rates and ice velocity (Cook et al., 2020; Bunce et al., 2021). Important processes at the ice surface are mainly described as air temperature-elevation and melt-albedo feedback. Air temperature and the large-scale and local circulation can change due to variations in surface altitude and surface reflectivity (albedo) as the surface changes with increasing melt from fresh snow to bare ice and finally to underlying solid ground (Vizcaino et al., 2015; Le clec'h et al., 2019; Ryan et al., 2019). Changes in air temperature again impact the ice properties creating a feedback loop.

These processes are in general addressed by regional climate models that include detailed parameterizations for albedo, surface melt and refreezing, runoff and snow drift and can provide ice sheet models with a total surface mass balance budget (Fettweis et al., 2020). Similarly, regional ocean models can provide highly detailed ocean thermal forcing or melt rates to ice sheet models to compute calving, frontal and sub-shelf melt rates (Zhao et al., 2022; Davis et al., 2023; Spears et al., 2023). In recent years, coupling regional atmosphere or ocean models with ice sheet models has been implemented to show the importance of interactive feedbacks from paleo to inter-annual time scales (Le clec'h et al., 2019; Zeitz et al., 2021; Pelle et al., 2021). However, these studies employ *regional* climate and ocean models and therefore lack the connections and feedbacks with the global climate system.

To investigate the global impacts of ice loss and potential feedbacks on the Greenland ice sheet, first steps have been done to integrate ice sheet models into CMIP models (Vizcaino et al., 2015; Lofverstrom et al., 2020; Ackermann et al., 2020; Muntjewerf et al., 2021; Smith et al., 2021; Madsen et al., 2022). Several simulations with the Earth System model CESM2 coupled to the ice model CISM2.1 have been performed with different greenhouse gas emission scenarios. These simulations overcome the resolution difference between global climate and ice sheet model by downscaling surface mass balance (Lipscomb et al., 2013; van Kampenhout et al., 2019; Sellevold et al., 2019). While their simulated Greenland ice mass loss is in agreement with other studies, the North Atlantic Meridional Overturning Circulation declines before the onset of increased Greenland ice sheet melt (Muntjewerf et al., 2020a, b; Lofverstrom et al., 2020; Muntjewerf et al., 2021). First simulations of a CMIP6 model with both the Greenland and Antarctic ice sheet under SSP1-1.9 and SSP5-8.5 scenario contribute to the feedback discussion by addressing increased basal melting under Antarctic ice sheets in combination with thickening grounded areas due to increase

of snow fall (Smith et al., 2021; Siahaan et al., 2022). These simulations show regional feedbacks but no clear impact on the global climate by the ice sheets.

Here, we present the first simulation with the Norwegian Earth System model coupled to an evolving Greenland ice sheet. Details and performance of the simulation setup are described in Goelzer et al. (2025). We evaluate a climate scenario from 1850 until 2300 from a historical forcing to an extended high emission scenario (SSP5-8.5) and compare results to the extended CMIP6 NorESM2 run with a steady Greenland ice sheet (NorESM2fixed). The main information for the coupling setting is described in section 2. Results and discussion are divided into general climate evolution within NorESM2 and the differences seen by including an evolving Greenland ice sheet simulated by CISM2.1 into NorESM2.

## 2  Model description and experimental setup

Our simulations are performed with the Norwegian Earth System model version 2 (NorESM2). They cover the historical and SSP5-8.5 forcing periods extended to 2300 and can be compared to the standard experiments of Seland et al. (2020) that were submitted to the Coupled Model Intercomparison Project phase 6 (CMIP6, Eyring et al., 2016). The model version is NorESM2-MM (Seland et al., 2020), equivalent to a nominal 1 degree resolution in atmosphere, land, ocean and sea-ice components resulting in a meridional resolution of 111 km over Greenland while the zonal resolution ranges from 13 to 55 km for atmosphere and land model. The component models are CAM6-Nor for the atmosphere (Kirkevåg et al., 2018; Bogenschutz et al., 2018), CLM5 for the land (Lawrence et al., 2019), the CICE5 sea-ice model of (Hunke et al., 2015), the river model MOSART (Li et al., 2013), the carbon-cycle model iHAMOCC (Tjiputra et al., 2020), BLOM as the ocean (Bentsen et al., 2013; Seland et al., 2020) and CISM2.1 as the land ice in Greenland (Lipscomb et al., 2019). Antarctica and other land ice areas are not changing in this setup. The sea level is held constant (volume conserving ocean) and there is no glacial isostatic adjustment of the bedrock.

The Community Ice Sheet Model (CISM v2.1,  Lipscomb et al., 2019) is a state of the art, parallel, 3-D thermomechanical ice flow model. Here, we use its standard configuration for Greenland: 4 km rectangular grid with 11 vertical levels and a depth-integrated higher-order approximation (DIVA) based on Goldberg (2011). Basal sliding is described via a pseudo-plastic sliding law combined with a simple basal hydrology model (Schoof and Hindmarsh, 2010; Aschwanden et al., 2013). This includes a spatially varying till friction angle dependent on bed elevation allowing lower yield stress at lower elevations resulting in a realistic velocity field for most Greenland (Aschwanden et al., 2016). Calving is implemented as a flotation criterion, removing ice when it becomes floating and routing this mass as water to the ocean.

For the CISM2.1 initialization, we first tune the basal friction parameters to match the observed present geometry. The ice sheet model spin-up is forced with pre-industrial NorESM2fixed SMB while keeping the outside margins to the present day extent. The spun-up ice sheet is included into NorESM2 as the initial 1800 geometry and NorESM2 is run together for 50 yrs to relax the climate system to the new ice configuration (including new albedo and freshwater fluxes). These steps are designed to keep initial climate conditions of NorESM2fixed and NorESM2 as close as possible facilitating comparisons of both evolving systems (Goelzer et al., 2025).

The coupling between CISM2.1 and the NorESM2 climate components is done at various time steps. The most frequent exchange happens between CLM and CISM2.1 providing CISM2.1 with annual surface mass balance (SMB) and ice surface temperature at the end of every NorESM2 year. SMB values are calculated on the current ice sheet surface elevation using 10 elevation classes to downscale SMB ranging from the 13 to 55 km global model grid to the 4 km ice sheet model resolution (Lipscomb et al., 2013; van Kampenhout et al., 2019; Sellevold et al., 2019; Muntjewerf et al., 2021) with the specified lapse

rate of $6.0\,°C\,km^{-1}$. The land model CLM experiences changes of ice area annually by adjusting the surface type (e.g. ice or rock) and thereby adjusting the energy exchange within NorESM2 to the current ice extent. The atmosphere (CAM) receives updated ice sheet elevation directly from CISM2.1 every 5 yrs through updates of topography and surface roughness. This 5 yrs coupling window is a compromise to keep the ice surface elevation updated and reduce the number of time-consuming recalculations of the restart files of CAM needed to set the new ice surface topography. For more information we refer to

(Goelzer et al., 2025).

In our setup, there is no direct exchange (i.e. thermal forcing and melting of ice in marine-terminating outlet glaciers) between the ocean model BLOM and CISM2.1. The fjords in Greenland with typically 2-10 km width are too narrow to be resolved by the ocean model and an additional downscaling or parameterization approach would be needed. Ice discharge and meltwater runoff (solid and liquid runoff) from the GrIS are passed to the ocean model. However, this solid and liquid runoff are calculated

in different ways for NorESM2 with evolving and fixed ice sheet.

In NorESM2 with an evolving ice sheet, snow and ice melt are calculated in the land model CLM and routed to the ocean as liquid runoff with the runoff scheme (MOSART). This melt is communicated as SMB forcing to the ice model CISM which is adjusting the ice mass of the GrIS accordingly. Ice areas that detach from the GrIS and ice loss due to calving are sent to BLOM as annually collected discharge. BLOM transfers this directly into a homogeneously freshwater input as a salinity

decrease in the closest corresponding ocean cell. The energy needed to melt the discharged ice is taken from the ocean heat reservoir. In NorESM2fixed, ice melt is also calculated within the land model, but the shape and hence the mass of the ice sheet stays the same during the entire simulation. Therefore, mass loss due to melted ice has to be compensated or distributed. In our setting, this mass loss is balanced by snow fall. Snow fall on a GrIS grid cell that reaches over 10 m is converted into solid ice. This ice mass gain - like the ice melt - can not be added to the GrIS mass and both terms are compensating each other.

Usually, the excess ice gain is distributed into the ocean as liquid runoff (like on NorESM2). However, in the case of extreme melt like in the end of our simulation, ice loss might be higher than ice gain. This artificially added ice again is taken from the ocean by creating a negative runoff in the nearest ocean cell to maintain the overall water balance of the coupled system. This different treatment is mediated for future NorESM2 simulations. For this publication however, we will refrain from discussing or comparing the runoff values since their definitions and calculations differ inherently. We therefore focus our discussion on

ice-atmosphere-land feedback and the overall impact on ocean circulation. We further acknowledge that NorESM2 itself is a system of coupled models similar to CESM2 (Danabasoglu et al., 2020). However, the focus of this study is the difference of NorESM2 with and without an evolving Greenland ice sheet. Hence, the use of the terminology "coupled" or "un-coupled" will in this document refer to NorESM2 (meaning NorESM2-CISM2.1) coupling versus the standalone NorESM2fixed simulation.

Overall, we have three simulations between 1850 and 2300 to compare: (1) NorESM2fixed: standard CMIP6 NorESM2-MM
setup (Seland et al., 2020) with a constant present-day Greenland ice sheet geometry, under an extended SSP5-8.5 scenario,
and (2) NorESM2: following setup (1) with an additional coupling to CISM2.1 and hence including an evolving Greenland
ice sheet. After 2100, both simulations are forced with $CO_2$ linearly reduced to less than $10\,\mathrm{GtCyr^{-1}}$ following scenarioMIP
(O'Neill et al., 2016) (Fig 1a). All other emissions are held constant at 2100 levels.

In addition, we can refer to a control simulation (3) an extended simulation with constant pre-industrial emission forcing
(280 ppm) from 1850 to 2300. In results and discussion, NorESM2 refers to the coupled NorESM2-CISM2.1 setup which has
the general characteristics as NorESM2fixed, like climate sensitivity. NorESMfixed emphasizes NorESM2 climate results with
a fixed Greenland ice sheet. For more details and performance, we refer to Goelzer et al. (2025).

## 3   Results

For the results, we will give a general overview of the mean global climate evolution from 1850 to 2300 focusing on differences
between the three simulations. This is followed by spatially varying global results displaying individual climate components
and the effects of the included Greenland ice sheet evolution. A few comparisons are done spatially for different times through-
out the simulations. For this, we use reference periods as: "1850s" as the mean of the first 20 years of simulations (1850-1870)
and average values of the end of the following centuries 1981 to 2000, 2081 to 2100, 2181 to 2200 and 2281 to 2300, named
"2000s", "2100s", "2200s" and "2300s" respectively. Whenever we refer to these periods, we refer to a 20 yrs mean.

### 3.1   Global mean changes

The global mean surface air temperature (SAT, at 2 m height above ground) fluctuates around 13.9°C for all three simulations
from 1850 to 2000, see Fig 1b. From 2000 onward, NorESM2 and NorESM2fixed show a steep increase in temperature until
2150 beyond which the rate of change decreases and the temperatures reaching 23.8°C by 2300 corresponding to a 10.0°C
anomaly compared to the 1850s. The trend in SAT is reflected in the ice sheet surface mass balance (SMB, Fig. 1c), which is
defined by how much snow is falling, how much is transformed into ice and how much ice is melting again per year provided
by the land module CLM. From the start of the simulation, the SMB is positive with a mean of $600\,\mathrm{Gt\,yr^{-1}}$. In 2000, the
SMB starts to diverge from the control run, reaching negative values in 2100 at a warming of 3.8 °C. At first, the SMB trend
is about -9 $\mathrm{Gt\,yr^{-1}}$. Between 2050 to 2100, the SMB change shifts to -29 $\mathrm{Gt\,yr^{-1}}$, between 2100 to 2200 and flattens first to
-19 $\mathrm{Gt\,yr^{-1}}$ in 2200 to 2250 and then -5 $\mathrm{Gt\,yr^{-1}}$ of change in the last 50 yrs of the simulation while showing high inter-annual
fluctuations. This leads to almost 1.47 m global mean sea-level contribution over the course of 250 yrs from the increase in
dynamic ice discharge and SMB decrease from 2050 onwards. The sea level contribution at the beginning of the simulation
period is influenced by the initialization to keep ice mass loss and margins close to observations and follows the control run
with pre-industrial forcing for the first decades.

For the ocean, global mean sea surface temperatures show a similar temporal evolution as surface air temperatures and (like
for SAT) NorESM2fixed and NorESM2 show the same trend. The mean sea surface salinity (Fig. 1e) diverges from the control

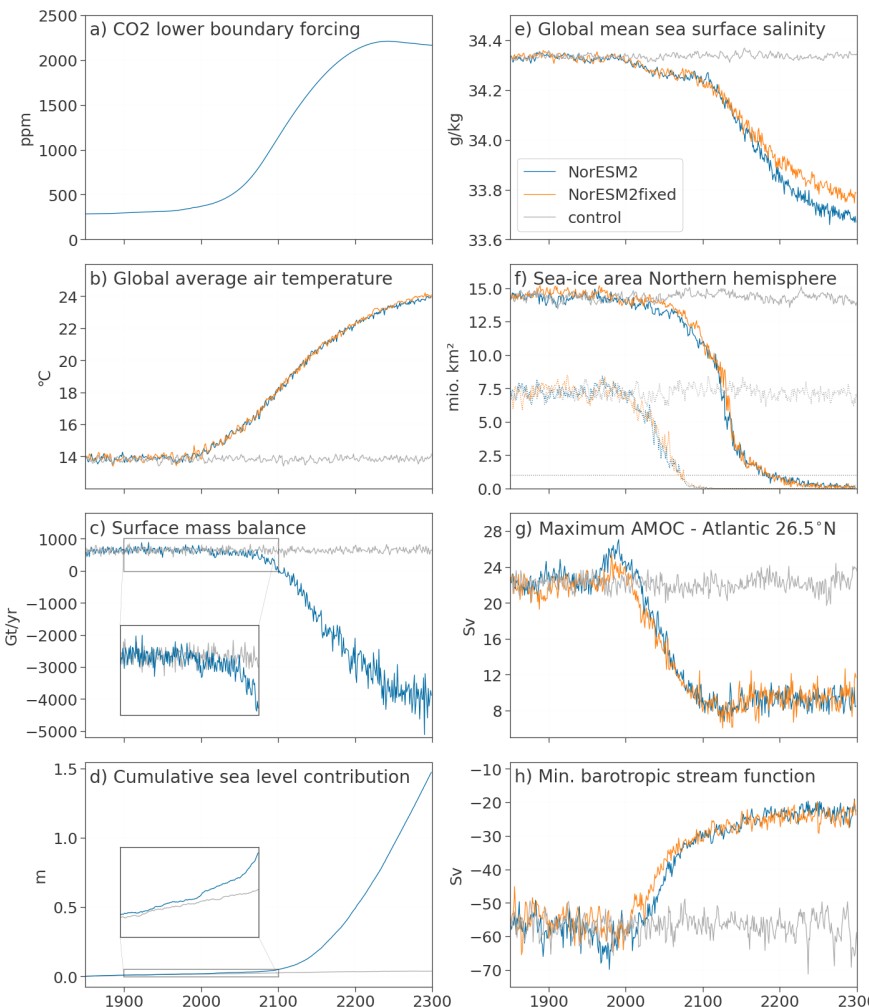

**Figure 1.** Evolution of global (a) $CO_2$ forcing (ppm) and resulting changes in global mean (b) surface air temperature (°C), (c) Greenland surface mass balance ($Gt\,yr^{-1}$), (d) Greenland ice sheet sea-level contribution (m), (e) sea surface salinity ($g\,kg^{-1}$), (f) area covered by sea ice for the northern hemisphere in August (dashed line) and February (solid line) ($m^2$) (g) maximum Atlantic meridional overturning circulation at 26.5°N (Sv), (h) minimum barotropic stream function around the Atlantic 70 to 20° N covering the sub-polar gyre. Control run with constant pre-industrial forcing is shown in gray, NorESM2 in blue and NorESM2fixed shown with orange lines.

run in both simulations around 2000 with a slow decrease from 34.3 to 34.2 $g\,kg^{-1}$ from 2000 to 2100. After 2150, the mean sea surface salinity in NorESM2 decreases faster than in NorESM2fixed. This results in a 0.1 $g\,kg^{-1}$ salinity difference in 2300 between NorESM2 and NorESM2fixed and an overall decrease in salinity of 0.7 $g\,kg^{-1}$ from 1850 to 2300.

Warming surface ocean water and air temperatures also induce sea ice changes. For the northern hemisphere, winter sea
ice extent decreases at the beginning of the 21st century slowly with increasing rates after 2050 and a steep drop in winter

**Table 1.** Global and Arctic (north of 60°N) surface air temperature changes since 1850 in 20 yr average periods as mean of the year (ANN), polar winter (DJF) and polar summer (JJA). Polar amplification factor is the ratio of temperature change in the Arctic divided by global change.

| | | 2100s | | 2200s | | 2300s | |
| --- | --- | --- | --- | --- | --- | --- | --- |
| | | NorESM2 | NorESM2fixed | NorESM2 | NorESM2fixed | NorESM2 | NorESM2fixed |
| global SAT change | ANN | 3.85 | 3.74 | 8.18 | 8.08 | 10.03 | 10.07 |
| | DJF | 4.06 | 3.92 | 8.44 | 8.32 | 10.27 | 10.30 |
| | JJA | 3.76 | 3.70 | 8.22 | 8.15 | 10.10 | 10.16 |
| arctic SAT change | ANN | 8.63 | 7.51 | 17.10 | 16.69 | 19.36 | 19.25 |
| | DJF | 12.21 | 10.58 | 22.73 | 22.35 | 25.22 | 25.09 |
| | JJA | 5.02 | 4.58 | 12.89 | 12.71 | 14.83 | 14.94 |
| arctic amplification | ANN | 2.24 | 2.01 | 2.09 | 2.07 | 1.93 | 1.91 |
| | DJF | 3.01 | 2.70 | 2.69 | 2.69 | 2.46 | 2.43 |
| | JJA | 1.34 | 1.24 | 1.57 | 1.56 | 1.47 | 1.47 |

coverage after 2120, leading to an all-year sea ice free ocean by 2200 (covering an area less than 1 mio. km$^2$). The maximum of the Atlantic meridional overturning circulation at 26.5°N (AMOC, Fig. 1g) is fluctuating between 20 and 27 Sv for the first 150 yrs. In this period, the control run shows the least fluctuations with values around 23 Sv. NorESM2 and NorESM2fixed show higher fluctuations, diverge in the first 50 yrs and increase with different magnitudes between 1950 and 2000. These differences might stem from different inter-decadal variability due to slightly different initial state. From 2000 onward, the AMOC decreases steeply for both NorESM2 and NorESM2fixed, stagnating by 2100. Note that the AMOC decline starts several decades before the decrease in SMB in NorESM2. After 2130, maximum AMOC values fluctuate around 10 Sv until the end of the simulation.

The barotropic stream function (defined as the depth-integrated volume transport of ocean water) can illustrate the position and strength of the subpolar gyre. This circulation system is known to react sensitively to freshwater forcing such as GrIS runoff (Born and Stocker, 2014; Swingedouw et al., 2021). The cyclonic circulation is represented with negative values in the barotropic stream function. Here, minimum values of the barotropic stream function (Fig. 1h) at the ocean area south-West of Greenland in NorESM2 and NorESM2fixed are diverging from the control run at the beginning of the 21st century, converging towards each other and fluctuating around -25 Sv from around 2170. This indicates a weakening of the subpolar gyre. NorESM2fixed shows weaker gyre circulation than NorESM2, but the difference is within inter-decadal variability.

In summary, the timing of change (meaning divergence from the control run) starts for each of the climate variables around 2000. For all of them, there are no major differences between NorESM2 and NorESM2fixed, except sea surface salinity.

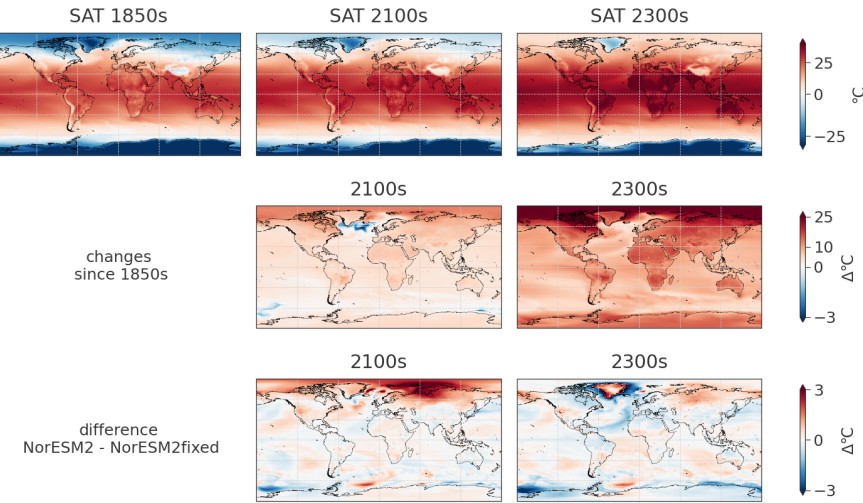

**Figure 2.** Surface air temperature evolution shown as absolute values 1850, 2100 and 2300 for NorESM2 (top row), changes since the start of the simulation for NorESM2 (second row) and as difference between NorESM2 and NorESM2fixed (third row). To cover differences of the initial climate state in NorESM2 and NorESM2fixed, the last row is calculated as double differences, meaning the difference of changes since the 1850s of the respective simulations.

## 3.2 Atmospheric changes

By 2100s, the annual average SAT increased over the oceans by up to 3 °C, between 5 to 8 °C over the continents and by about 12 °C over the Arctic Ocean (Fig. 2, Fig. A1). The only exception to the warming trend is the area southeast of Greenland and south of Iceland where SAT decreased by 4 °C. By 2200s, SAT are above freezing point everywhere except over Greenland and Antarctica. These changes continue until reaching an increase of 10 °C over the oceans, 15 °C over the continents and over 25 °C in the Arctic ocean, Northern Canada and Siberia by the end of the simulation.

This evolution is in general similar for NorESM2fixed and NorESM2. However, NorESM2 shows an additional warming over Siberia and the neighboring ocean area by 2100s with about 3 °C difference compared to NorESM2fixed. In 2200s, NorESM2 is about 1 °C warmer than NorESM2fixed in the entire Arctic except the Greenlandic coast as well as Antarctica, the Indian Ocean and the Southern Atlantic Ocean. By 2300s, these differences disappeared, showing even an increased warming over the Antarctic in NorESM2fixed compared to NorESM2. Around the coastline of Greenland however, the SAT in NorESM2 are increasingly colder than in NorESM2fixed while over Greenland we see the opposite showing the local surface air temperature changes due to the Greenland ice sheet shrinking. The difference in surface air temperature stems from the additional freshwater influx around the coast of Greenland which increases ocean stratification and reduces vertical heat exchange leading to surface cooling. This does not explain the initial lack of melting which is due to the cold initial bias (discussed in Goelzer et al. (2025)).

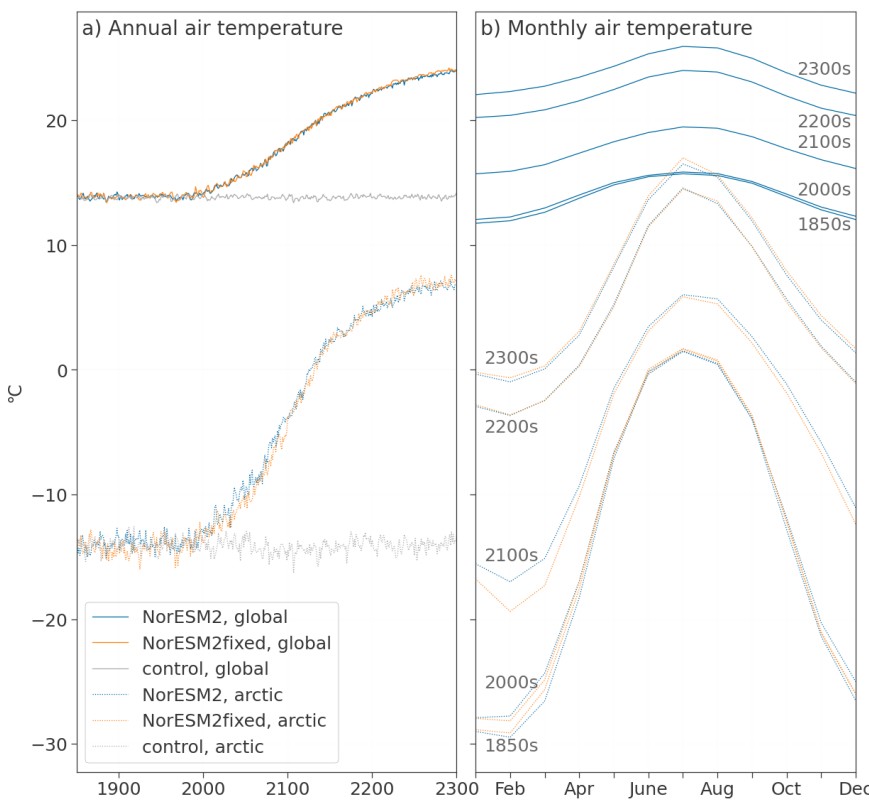

**Figure 3.** Mean global (solid lines) and Arctic (dotted lines) surface air temperatures as (a) annual mean temperatures over the entire simulation time and (b) as monthly means over a 20 yr period in 1850s, 2000s, 2100s, 2200s and 2300s. NorESM2 in blue, NorESM2fixed in orange, control in gray. Temperatures are given in °C.

Also seasonally, SAT increase more in the Arctic compared to global temperatures from 1850s to 2300s (Fig. 3, Table 1, Fig. A2, Fig. A3). Arctic summer (June-July-August: JJA) temperatures increase on average by 14.8 °C (from 3.1 to 17.9 °C) and in winter (December-January-February: DJF) by 25.1 °C (from -25.2 to 0.0 °C). Globally, JJA and DJF temperatures increase by 10.2 and 10.3 °C respectively.

This change happens mainly between 2000s and 2200s with global mean SAT increase by 0.1 °C, 3.8 °C, 4.3 °C and 1.9 °C between 1850s and 2000s, and the next centuries respectively. For the Arctic, the biggest steps in SAT increase happen as well between 2000s and 2200s with changes of -0.4 °C, 5.4 °C, 7.8 °C and 1.9 °C over the arctic summers (JJA) of each century and with 0.9 °C, 11.3 °C, 10.5 °C and 2.5 °C over winter (DJF) during the same time periods (Fig. 3, Table1). The NorESM2fixed simulation shows the same seasonal evolution patterns except for the Arctic SAT in 2100s where DJF SAT are 1.6 °C below NorESM2 and 0.3 °C below in JJA. By 2200s, SAT reach the same level in NorESM2fixed and NorESM2. The difference in global and Arctic temperature changes is usually expressed by Arctic (or polar) amplification, e.g. the ratio of changes of Arctic SAT versus global SAT. Polar amplification together with climate sensitivity of the model is an important control on the

Greenland ice sheet mass loss within a simulation and shows how quickly the Greenland ice sheet mass changes are impacted by $CO_2$ increase (Fyke et al., 2014; Hofer et al., 2020; Xie et al., 2022). The Arctic amplification by 2100s in NorESM2 is with 2.2 larger than in NorESM2fixed (2.0). By 2300s however, values are almost the same with 1.91 and 1.93 (Table 1).

The pattern of global annual precipitation is largely unchanging over the entire simulation period but the magnitude changes
(Fig. 4). The strong precipitation bands over the equatorial ocean are increasing already from the 21st century onward and reaching values of maximal $20\,\mathrm{mm\,d^{-1}}$ in 2300s (changes up to $7\,\mathrm{mm\,d^{-1}}$). The northern hemisphere and Antarctica show increasing precipitation especially along coastal regions. While Central America and the northern part of South America, Africa and Oceania see a decrease in precipitation by up to $-3\,\mathrm{mm\,d^{-1}}$ by 2300.The differences between NorESM2fixed and NorESM2 are minor with max. $\pm1.5$ mm, but they are visible since the start of the simulation in 1850 and persistent, hence, negligible and
due to slightly different initialization states of NorESM2 and NorESM2fixed. The increase in annual precipitation is mainly during DJF. Precipitation in JJA is reduced except along the South and East coast of Asia and Antarctica (Fig. A4), Fig. A5. DJF snowfall, however, first increases over the Nordic Sea, northern North America, Greenland and Siberia while decreasing over southern North America, the Atlantic, Europe and the Bering Sea (Fig. A6). During the southern winter (JJA), snowfall shifts from the Southern Ocean towards Antarctica, leading to increased snowfall along the Antarctic coast and a decrease over
the Southern ocean close to the coast (Fig. A7). The Arctic winter snowfall (DJF) in NorESM2 is higher (up to $0.5\,\mathrm{mm\,day^{-1}}$) in Scandinavia, East Greenland, parts of the Nordic Sea and North-East Canada compared to NorESM2fixed in 2300.

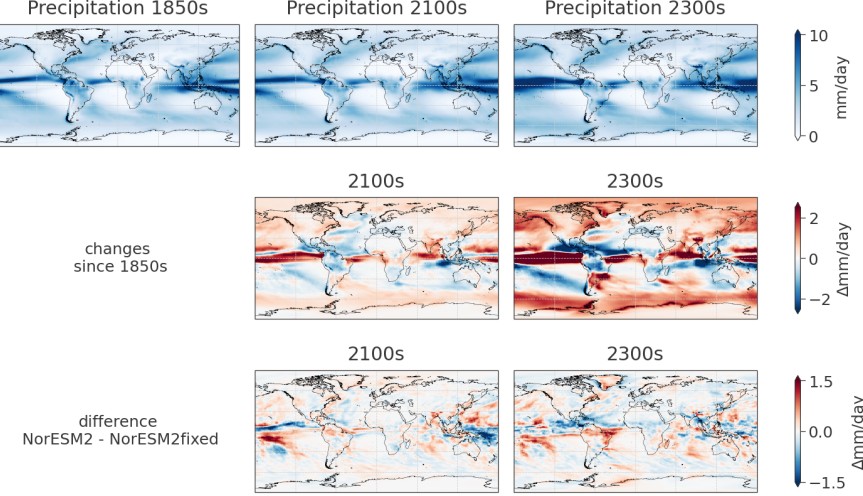

**Figure 4.** Total annual precipitation with absolute values 1850, 2100 and 2300 for NorESM2 (top row), changes since the start of the simulation for NorESM2 (second row) and as difference between NorESM2 and NorESM2fixed (third row). To cover differences of the initial climate state in NorESM2 and NorESM2fixed, the last row is calculated as double differences, meaning the difference of changes since the 1850s of the respective simulations.

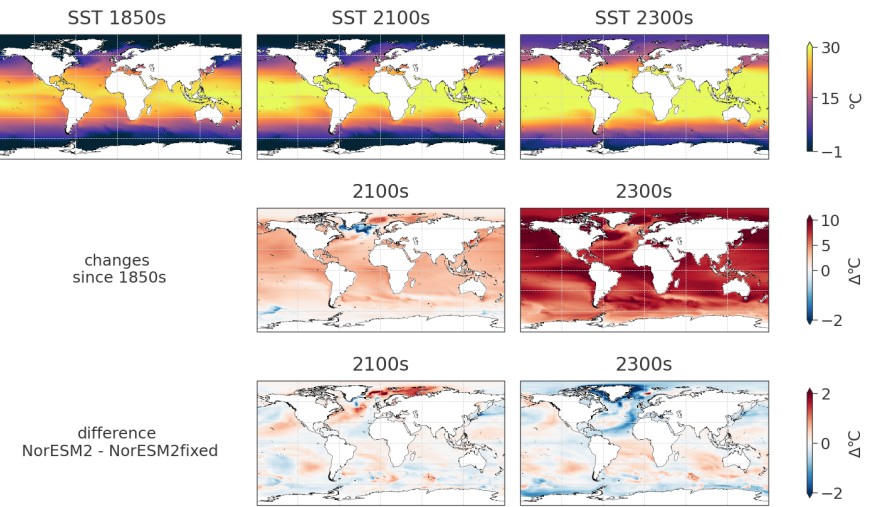

**Figure 5.** Mean annual sea surface temperature evolution shown as absolute values 1850s, 2100s and 2300s for NorESM2 (top row), changes since the start of the simulation for NorESM2 (second row) and as difference between NorESM2 and NorESM2fixed (third row). To cover differences of the initial climate state in NorESM2 and NorESM2fixed, we show double differences here, meaning the difference of the anomaly in 2100s (and 2300s) for NorESM2 and NorESM2fixed.

### 3.3 Oceanic changes

At the beginning of the simulation, annual mean sea surface temperatures spread from negative values around the pole, increasing towards the equator to around 35 °C (Fig. 5). By the end of the 20th century, sea surface temperatures (SSTs) increase by 1 to 5 °C in both NorESM2fixed and NorESM2. One exception is the eastern subpolar North Atlantic (Fig. 5). The cooling in the Nordic Seas, between Greenland and Scandinavia, is offset by an increased warming area between Iceland and Svalbard which has been predicted and discussed often (Drijfhout et al., 2012; Caesar et al., 2018). Both effects (local warming and cooling) are more pronounced in NorESM2 than in NorESM2fixed. By the 2300s, the SSTs increase by an average of 7.5 °C since 1850s and with maximum warming of up to 14.7 °C in the Labrador Sea, the Greenland Sea, Barent Sea and the North Pacific Ocean. The area south of Greenland shows almost no change in NorESM2 while warming is minimal in NorESM2fixed. By the end of 2200, NorESM2fixed and NorESM2 start to show differences in surface ocean temperatures around Greenland with on average 2 °C colder ocean around Greenland in NorESM2.

For surface salinity, projected changes are more pronounced regionally (Fig. 6, Fig. A8). The North Atlantic (above 50°N) and the Arctic Ocean are becoming fresher in both simulations with surface salinity changes between -1.5 and 0 g kg$^{-1}$ by 2100 and -5 g kg$^{-1}$ to -2 g kg$^{-1}$ by 2300. Differences in the coupled and uncoupled simulation are visible in a fresher Davis Strait and Baffin Bay in the coupled run by 2100 spreading later-on to around Greenland reflecting the additional freshwater provided by a changing GrIS within NorESM2. The coastal areas of North America and Asia are more saline in the coupled run. By 2100, we see an increasing meridional gradient in the Atlantic. The fresher areas in the North become even fresher and

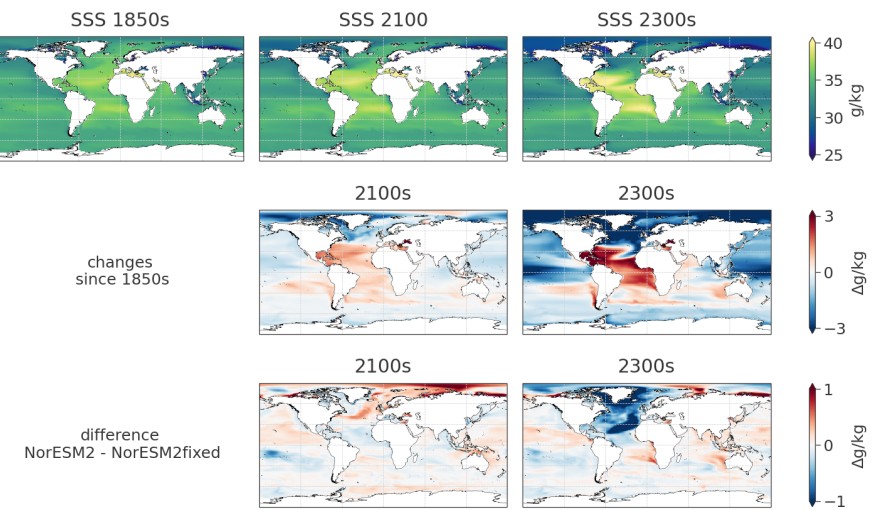

**Figure 6.** Sea surface salinity evolution shown as absolute values 1850s, 2100s and 2300s for NorESM2 (top row), changes since the start of the simulation for NorESM2 (second row) and as difference between NorESM2 and NorESM2fixed (third row). To cover differences of the initial climate state in NorESM2 and NorESM2fixed, we show double differences here, meaning the difference of the anomaly in 2100s (and 2300s) for NorESM2 and NorESM2fixed.

areas below 40°N become even more saline throughout time. This is valid for both simulations, however the entire Atlantic in the northern hemisphere is fresher with an evolving GrIS. In deeper layers (100–500 m depth), salinity does not change as significantly. However, there is an area of high salinity building in the Atlantic between 0 and 50°N peaking around 400 m depth and being highest in the Caribbean Sea towards the open Atlantic Ocean (Fig. A9). This pattern matches the increased temperatures in the same region.

### 3.4 Sea-ice evolution

The sea ice shows a very similar evolution for both simulations. In 1850s conditions, the maximum extent stretches from the coast of North America, starting in Baffin island, around Greenland, northern Iceland, covering Svalbard and the northern coast of Siberia, to parts of the Bering Sea and the Sea of Okhotsk. The 1850s minimum sea ice extent is covering a large area as well, from Davis Strait over the entire Arctic Ocean to Bering Strait. By 2100, the maximum sea ice extent retreated in all surrounding bays, but still covers the main Arctic ocean and its surrounding coasts. Even though changes in the maximum sea ice extent are small, the summer sea ice extent is already shrinking starting in the middle of the 21st century. Hence in 2100, summer sea ice extent is covering primarily the Arctic ocean around the Queen Elizabeth Islands and northern Greenland between the North Pole around 75°N. By 2080, the Arctic ocean is ice free during summer and all-year ice free by 2200 (see Fig. 1) which means sea-ice coverage is in an area less than 1 million km$^2$ for both simulations. The small remaining winter

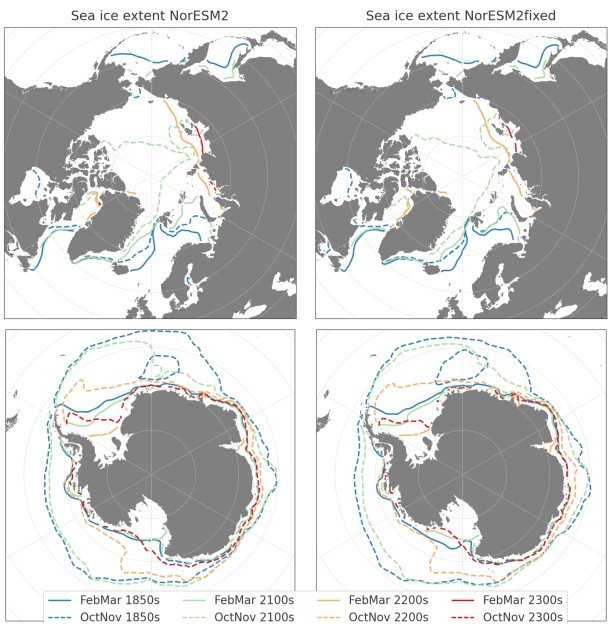

**Figure 7.** Contour of minimum and maximum sea ice extent covering at least 15% of the grid cell for northern and southern hemisphere. Northern hemisphere minimum extent (solid line) is taken as the mean of October/November sea ice coverage and maximum extent (dashed line) is February/March coverage over 20 yrs in the given period. For southern hemisphere solid and dashed lines mark the same time and show maximum and minimum extend respectively.

sea ice after 2200 stays around the coast of North Greenland/Baffin Bay and on the other side around the coastline of Laptev and Kara Sea.

## 3.5 Greenland ice-sheet changes

Changes on the Greenland ice sheet in NorESM2 are minor before the 2000s and only visible around the ice margins (Fig. 8 and Fig. A10). Spatially, the SMB is mainly positive with a narrow ablation area covering solely 2% of ice area and hence an overall mass gain for the ice sheet over the 20 yrs average is recorded. The regions with most pronounced ablation areas (negative SMB, where ice melt is higher than snowfall) are along the west coast of Greenland, visible already in the 2000s, increasing and spreading along the coast until 2100. By this time, additional melt areas start to appear along the North and North-East coast. During the transition from 2100s to 2200s, the ablation area increases from 13% to 69% of the ice area. By 2200s, melt is dominant around all ice margins of the entire GrIS and spreads far inland. The accumulation area is limited to above 2250 m with a maximum annual mass gain of 7.5 m yr$^{-1}$ locally. In 2300s, 93% of the Greenland ice sheet experiences mass loss with values up to -12 m yr$^{-1}$ locally.

Similarly, ice thickness and velocity change first around the margins with slowly increasing changes towards the interior of the ice sheet (Fig. 8). Changes are most pronounced around the west and north coasts spreading over time around the entire

**Table 2.** Surface mass balance changes since 1850 (Gt yr$^{-1}$), separated into contributions from climate (SMB$_{clim}$), elevation (SMB$_{elev}$), and the total.

|  | SMB$_{clim}$ | SMB$_{elev}$ | SMB |
| --- | --- | --- | --- |
| 2100s total | -356.8 (104%) | 14.8 (-4%) | -342.0 |
| accumulation | 94.4 | 51.1 | 93.5 |
| ablation | -451.2 | -36.4 | -435.5 |
| 2200s total | -2527.9 (79%) | -659.7 (21%) | -3187.7 |
| accumulation | 15.8 | 233.0 | 21.6 |
| ablation | -2543.7 | -892.7 | -3209.3 |
| 2300s total | -2269.6 (51%) | -2223.0 (49%) | -4492.5 |
| accumulation | 0.0 | 180.4 | 0.1 |
| ablation | -2269.6 | -2403.4 | -4492.6 |

coastline and further inland. Ice surface velocities increase by up to 600m yr$^{-1}$ for lower elevation areas of marine-terminating glaciers. Decrease in velocity by up to 100 m yr$^{-1}$ is simulated for higher elevation areas where ice thickness is large, ice surface slope is low and SMB is becoming negative after 2200s.

As explained in section 2 (Model description and experimental setup), direct SMB comparison between the two simulations is not possible due to a different treatment within the land module CLM when keeping the ice areas fixed. This has been improved and future simulations can be better compared when running with and without an evolving Greenland ice sheet. In order to compare the effects of the changing climate versus the changing ice surface elevation, we use the elevation classes within NorESM2 to calculate SMB in 2100s, 2200s, 2300s on the 1850s ice surface elevation. This SMB$_{clim}$ represents the part of the SMB only due to the changing climate since we convert it into a system with fixed GrIS surface elevation. The residual of the actual SMB and this SMB$_{clim}$ is expressing the SMB$_{elev}$ due to the elevation feedback, including changing temperatures and precipitation alike. We acknowledge this to be a simple approximation because climate and elevation changes are impacting each other. Yet, this is a way to show where and when the simulated elevation changes are substantial due to the missing data output of SMB and elevation classes of NorESM2fixed.

In 2100s, there is up to 200 mm yr$^{-1}$ more snow fall in the accumulation area and over -2 m yr$^{-1}$ more ablation due to the climate compared to 1850s. The elevation change leads to 200 mm yr$^{-1}$ accumulation in the North in the area where there's increased ablation due to climate. Further, there's a band of increased ablation (up to 500 mm yr$^{-1}$) neighboring a band of increased accumulation, showing the shift in melt and snowfall line due to the thinning of the ice sheet (Fig. 8, Fig. A10).

By 2200s, the accumulation area retreats to above 2800 m elevation and ablation around the margins increases to -2 to -5 m yr$^{-1}$ due to climate compared to 1850s. The increased snow fall band due the the elevation change wanders from the North

coast 150 to 300 km inland. Ablation due to elevation changes is more pronounced represented by a wide-spread band around the coast up to 200 km inland of about -1 to -2 m yr$^{-1}$ compared to 1850s.

In 2300s, there is a decrease of accumulation all over the GrIS due to climate compared to 1850s. The ablation increased to up to -5 m yr$^{-1}$ and spreads further inland. While the ablation due to climate spreads more inland, it is less pronounced and smaller than ablation due to elevation around the ice margins. There are still small areas of increased snow fall of up to 100 mm yr$^{-1}$ above 2800 m due to elevation changes. This is however surpassed by the melt increase of SMB$_{clim}$ in this area.

    In 2100s, the warming climate dominates the SMB changes and ice surface elevation changes are minor until this point and

295 contribute as SMB increase to the total SMB change (Table 2). In 2200s, the elevation changes are amplifying the melt around the ice margins and also introduces increased snow fall in the North. From 2200s to 2300s, the melt due to elevation feedback overtakes the melt amount due to climate around the ice margins. While in 2200, the elevation feedback contributes 21% to the negative SMB, in 2300 SMB$_{elev}$ is gaining on impact and contributes to almost 50% of the SMB. At the end of the simulation, climate and elevation-feedback play an almost equally important role for the SMB. This shows the importance of coupling ice

and climate models after 2100. For more detailed evaluation of the historic surface mass balance we refer to Goelzer et al. (2025).

## 4   Discussion

This study compares the climate evolution of NorESM2 with a steady and evolving Greenland ice sheet. Over 450 yrs of simulation and a CO$_2$ increase from 280 ppm to 2100 ppm following SSP5-8.5 and its extension (O'Neill et al., 2016), NorESM2

simulates a warming of 3.75 and 10.03 °C and 0.05 m and 1.47 m sea-level contribution from the GrIS in 2100 and 2300 respectively.

In NorESM2, the AMOC is slowing down before the onset of increased GrIS mass loss. The Arctic sea ice is shrinking earlier within NorESM2, but the NorESM2fixed simulation catches up after 50 yrs with the same retreat rates onward. The Arctic amplification is factor 0.2 (almost 1°C more) higher in NorESM2 than in NorESM2fixed around 2100 due to higher winter

temperatures over the Arctic Ocean and Siberia. In 2200 however, Arctic amplification is almost the same in both simulations. Overall, simulations with and without a changing GrIS are similar. The global climate evolution is dominated by the extended high emission scenario and the effect of the evolving GrIS is negligible in our setting. For local climate around the GrIS, however, the exchange between atmosphere, land and ice model lead to a difference in climate around the ice margins. Further, ice elevation changes lead to changes in precipitation (snow and rain fall alike) and increased melt compared to a fixed ice

geometry and are important for the GrIS mass loss projections after 2100.

    The simulated ice loss from the Greenland ice sheet is on the lower end compared to other models. For example, in the ice sheet model intercomparison project for CMIP6 (ISMIP6, Goelzer et al., 2018, 2020), various ice sheet models are forced with atmospheric and ocean forcing from a range of CMIP5 models with parameterized SMB-height feedback, but without two-way coupling to the climate models. Simulations from the full ISMIP6 set project on average a sea-level contributions 90±50 mm

(2σ range) by 2100. The ensemble of ISMIP6 ice sheet models forced with output from NorESM1, the CMIP5 version of our

climate model, gives 69±38 mm sea-level contribution under RCP8.5 forcing by 2100, compared to 48 mm in our setup. Extensions beyond 2100 of RCP8.5/SSP5-8.5 are performed in different ways and are hence harder to compare. We can however say that the result from different methods show a wide range with sea-level contributions between 0.97 m and 3.74 m by 2300 under RCP8.5 (Aschwanden et al., 2019; Zeitz et al., 2021; Beckmann and Winkelmann, 2023), with our results of

1.47 m in the lower third. The lack of calving mechanism and direct ice-ocean interaction might underestimate dynamic ice loss due to the warming ocean and increased surface runoff. It is unclear which of these factors is the biggest contributor to the low end, but together with the high warming, the result of low Greenland ice sheet impact on global and regional climate has to be taken carefully.

Underestimating GrIS mass loss and its impact on the climate system is also suggested for the simulations by Muntjewerf

et al. (2020b). Even though SMB is projected to be the main driver for Greenland's mass loss, ocean-induced retreat and resulting dynamical mass loss is an important factor for its changes (Haubner et al., 2018; Rahlves et al., 2024). There is already work in progress to implement observational-trained prescribed retreat following ISMIP6 (Slater et al., 2019) into our model (Rahlves et al., 2024). Other options would be to decide on a calving scheme (defining how much ice breaks off as icebergs) independent of fjord temperatures, since those are not resolved in NorESM2. However, a uniform calving law that

predicts tidewater glacier retreat for all glaciers around Greenland has not been agreed upon yet (Benn et al., 2017; Choi et al., 2018; Amaral et al., 2020).

Our simulation is in agreement with the emerging pattern for CMIP6 models to simulate 30 to 50% of AMOC decline by 2100 which is independent of the climate scenario at least up to 2060 (Weijer et al., 2020; Baker et al., 2023). We cannot confirm any direct influence from GrIS mass loss to the further AMOC decline in our setting since the AMOC evolution of

NorESM2fixed and NorESM2 is very similar from 2050 onward. The GrIS mass loss and freshwater input to the ocean appears to not be large enough to substantially affect the AMOC beyond its response to other drivers.

Integrating active ice sheets into CMIP6 models is still under development. UKESM1.0 publications include both the Greenland and Antarctic ice sheet, running solely to 2100 and focusing discussion on the Antarctic ice sheet results on a lower emission scenario (SSP1-1.9) (Smith et al., 2021; Siahaan et al., 2022). The published EC-Earth setup is based on CMIP5 and

their SMB is coarse resolution and leads to a third of melt compared to CESM2 in a doubled $CO_2$ scenario (Madsen et al., 2022). Hence we refrain from comparison as well. For comparison with other CMIP6 models integrating an ice sheet model, we will focus our discussion on CESM2.1-CISM2.1 (Muntjewerf et al., 2020a, b). CESM2.1-CISM2.1 is an Earth system model similar in most modules to NorESM2 and the same ice sheet model and we will focus on two of their many simulations done with an integrated Greenland ice sheet. CESM2.1-CISM2.1 GrIS contribution to SLR by 2100 in the SSP5-8.5 scenario

is 109 mm (Muntjewerf et al., 2020a) which is more than double NorESM2's contribution (48.4 mm). A longer simulation is done with a lower $CO_2$ increase. It shows the same pattern of SMB, velocity and thickness change in 2100 (after 140 yrs of simulation) as in (Muntjewerf et al., 2020a) even though $CO_2$ values are lower (Muntjewerf et al., 2020b). Hence, we can try to carefully compare NorESM2 runs with the simulation discussed in Muntjewerf et al. (2020b). In CESM2.1-CISM2.1, the polar amplification factor in 2100 is 1.6 for the Arctic (north of 60°N; Muntjewerf et al., 2020b), compared to 2.2 in our simulation

(Table 1). By 2300s, there is 1.1 m sea-level contribution in CESM2.1-CISM2.1 and 8.5 °C global warming (Muntjewerf et al.,

2020b) compared to 1.5 m sea-level contribution and 10.0 °C warming in NorESM2. Meaning, even with almost double the $CO_2$ values, NorESM2 simulates only a small increase in mean global average SAT and global sea-level increase compared to CESM2.

The ice sheet in CESM2.1-CISM2.1 is initialized to evolve freely and with a long spin-up forced by past climate and while in NorESM2 it is constrained to match present day ice extent. This leads to large differences in initial ice volume and hence in distinct potential ice loss. In particular, CESM2.1-CISM2.1 has thicker and further extended ice sheet margins that are ready to melt in a warming climate. Even though the ocean modules are different in NorESM2 and CESM2, both simulations find an AMOC decrease before substantial Greenland ice sheet mass loss (Muntjewerf et al., 2020b). In another setup, a long-term CESM2.1-CISM2.1 simulation over the last interglacial (Sommers et al., 2021) presents a negligible effect of increased GrIS freshwater to the AMOC, too. These findings are in contrast to Earth system models of intermediate complexity (EMICs) with integrated Greenland and Antarctic ice sheet. There, freshwater fluxes from the ice sheets have a mitigating effect on changes in the climate system and impact both ocean circulation and air temperatures (Goelzer et al., 2011; Li et al., 2024). This is interesting and likely due to differences in ocean and atmospheric circulation which are strongly simplified within EMICs compared to CMIP models (like NorESM2 and CESM2.1).

Climate sensitivity is a measure for changes in simulated global air temperature in response to increased $CO_2$ forcing. There are different metrics used within the CMIP scheme to define climate sensitivity and NorESM2 is on the lower end of the commonly definitions used compared to other CMIP6 models (Seland et al., 2020; Gjermundsen et al., 2021). For example, NorESM2 is with a mean global SAT increase of 3.6 °C from 2015 to 2100 on the lower end compared to the CMIP model mean of 5 °C and raging from 3.5 to 7 °C under RCP8.5/SSP585. This also shows in the comparison with CESM2.1-CISM2.1 and ISMIP6 results discussed above. Combined with Arctic amplification, climate sensitivity is used as an indicator for Greenland ice sheet changes (Fyke et al., 2014). Low GrIS mass changes allude to a low Arctic amplification and climate sensitivity within NorESM2. With low climate sensitivity, we also expect less impact of the Greenland ice sheet into the global climate system (Goelzer et al., 2011). On top of this, we have no direct ocean-ice interaction enforced in our simulation setup. Hence, results here might be comparable with lower emission scenarios, even though we simulate under the SSP5-8.5 scenario. This emphasizes the importance to put results of coupled systems of climate-ice sheet model into perspective of their climate sensitivity. Further, this work nicely adds to the lower spectrum compared to CESM2 and UKESM1.0 simulations with high climate sensitivities.

## 5 Conclusions

Here, we analyse the first NorESM2 simulation with an evolving Greenland ice sheet. This long-term simulation gives the possibility to assess trends and impacts compared to the standard NorESM2 setup with a fixed ice sheet. While global warming under the high emission SSP5-8.5 scenario dominates the changes for most climate components, we see some differences in Arctic temperatures and sea surface salinity around 2100. However these differences are negligible by the end of the simulation in 2300. Further, the feedback between climate and ice sheet due to ice surface elevation changes becomes important after 2100.

From 2200 onward, these elevation feedbacks are making up for almost half of the SMB change what makes them a major contributor to the decreasing surface mass balance. We therefore encourage simulations beyond 2100 or 2200 if the effort is taken to include an ice sheet into climate simulations and recommend accommodating elevation feedbacks into climate models from at least 2100 onward.

Climate sensitivity is a measure for changes in simulated global air temperature changes in response to increased CO2 forcing. NorESM2 often occupies the lower end of the range set by CMIP6 models (Seland et al., 2020; Gjermundsen et al., 2021). The comparatively low climate sensitivity of NorESM2 is also evident in comparison with the very similar CESM2.1-CISM2.1 and may explain our model's performance with regard to the ISMIP6 results discussed above. Independent of the climate sensitivity, changes on the Greenland ice sheet so far have none or minor impact on the global system for CMIP6 models - which contrasts model studies with EMICs.

More work has to be done to integrate ice-ocean interactions at the calving front and below ice shelves to further include the Antarctic ice sheet into NorESM2. Until this is done, there is added value in the work presented here by focusing at only the GrIS and hence see signals due to this coupling can stem only from this ice sheet. This is a big step into more integrated ice sheets into climate models and NorESM2 will continue to participate in intercomparison projects under CMIP7 in the future.

*Code and data availability.* The NorESM2 model code is developed and freely available under https://github.com/NorESMhub/NorESM. The specific code repository used to set up the model is archived under https://doi.org/10.5281/zenodo.11199967. The full code used to produce the coupled experiments is persistently archived under https://doi.org/10.5281/zenodo.11200059. The raw data for the coupled NorESM2 experiments has been archived with persistent identifiers https://doi.org/10.11582/2024.00079, https://doi.org/10.11582/2024.00080, https://doi.org/10.11582/2024.00081 and https://doi.org/10.11582/2024.00082. The CMORized output from the NorESM2fixed experiments (Bentsen et al., 2019a;b) can be accessed through the ESGF at https://doi.org/10.22033/ESGF/CMIP6.8040 and https://doi.org/10.22033/ESGF/CMIP6.8321.

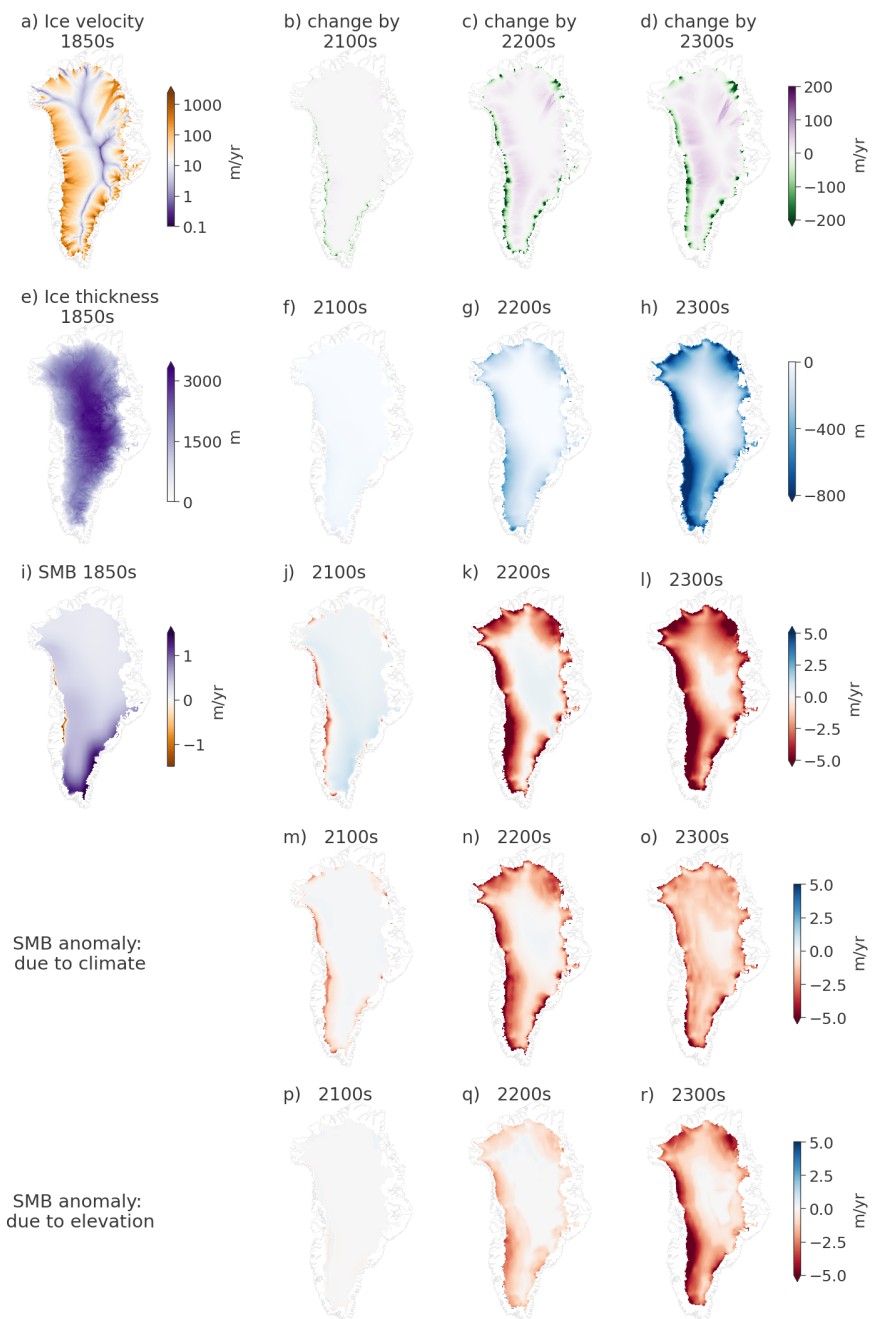

**Figure 8.** Overview of spatial GrIS changes within NorESM2. (a) Initial ice surface velocity ($\mathrm{m\,yr^{-1}}$) in 1850s and as velocity changes in (b) 2100s, (c) 2200s and (d) 2300s since 1850s. (e) Initial ice thickness (m) in 1850s and as thickness changes in (f) 2100s, (g) 2200s and (h) 2300s since 1850s. (i) Surface mass balance ($\mathrm{m\,yr^{-1}}$) in 1850s and as anomaly in (j) 2100s, (k) 2200s and (l) 2300s to 1850s. Contribution of climate and elevation to SMB anomaly in (m),(p) 2100s, (n),(q) 2200s and (o),(r) 2300s respectively. Gray outline marks the coast line of Greenland.

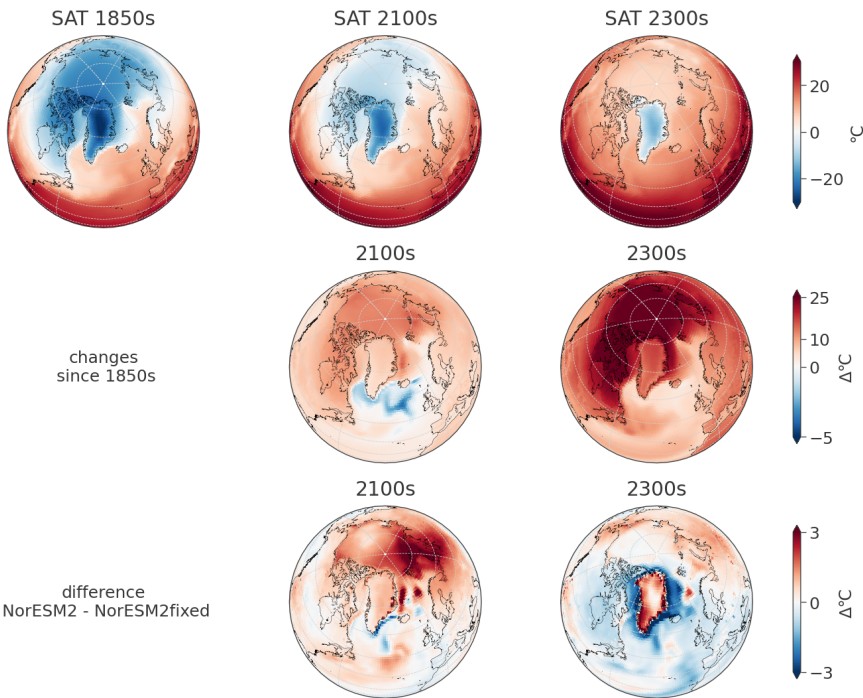

**Figure A1.** Annual surface air temperature as shown in Fig. 2 on Arctic projection. Absolute values 1850, 2100 and 2300 for NorESM2 (top row), changes since the start of the simulation for NorESM2 (second row) and as difference between NorESM2 and NorESM2fixed (third row). To cover differences of the initial climate state in NorESM2 and NorESM2fixed, the last row is calculated as double differences, meaning the difference of changes since the 1850s of the respective simulations.

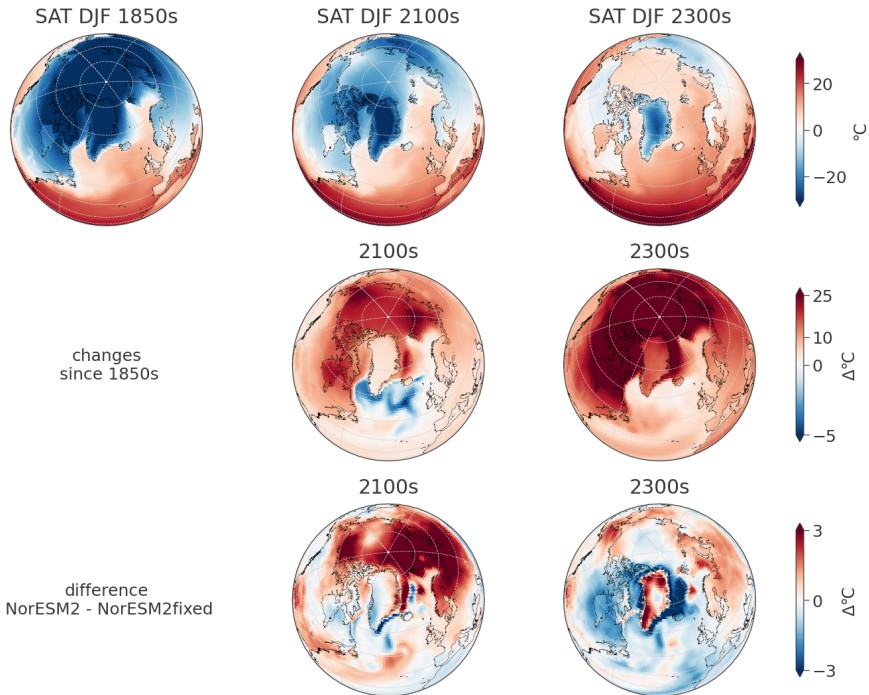

**Figure A2.** Arctic winter (DJF) surface air temperature. Absolute values 1850, 2100 and 2300 for NorESM2 (top row), changes since the start of the simulation for NorESM2 (second row) and as difference between NorESM2 and NorESM2fixed (third row). To cover differences of the initial climate state in NorESM2 and NorESM2fixed, the last row is calculated as double differences, meaning the difference of changes since the 1850s of the respective simulations.

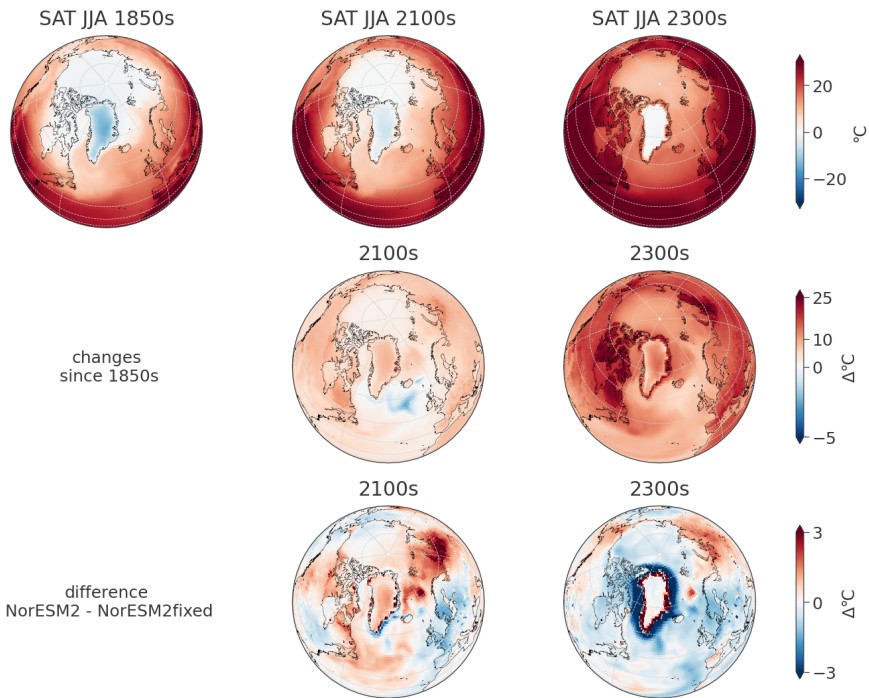

**Figure A3.** Arctic summer (JJA) surface air temperature as shown in Fig. 2 on Arctic projection. Absolute values 1850, 2100 and 2300 for NorESM2 (top row), changes since the start of the simulation for NorESM2 (second row) and as difference between NorESM2 and NorESM2fixed (third row). To cover differences of the initial climate state in NorESM2 and NorESM2fixed, the last row is calculated as double differences, meaning the difference of changes since the 1850s of the respective simulations.

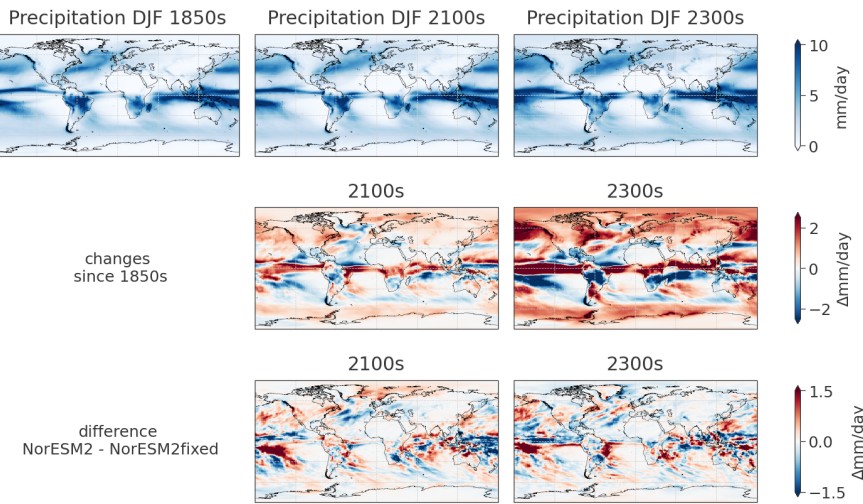

**Figure A4.** DJF precipitation with absolute values 1850, 2100 and 2300 for NorESM2 (top row), changes since the start of the simulation for NorESM2 (second row) and as difference between NorESM2 and NorESM2fixed (third row). To cover differences of the initial climate state in NorESM2 and NorESM2fixed, the last row is calculated as double differences, meaning the difference of changes since the 1850s of the respective simulations.

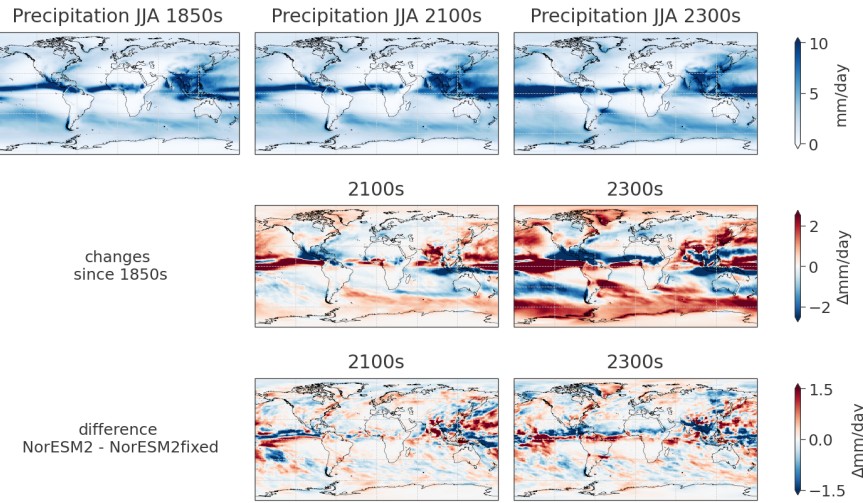

**Figure A5.** JJA precipitation with absolute values 1850, 2100 and 2300 for NorESM2 (top row), changes since the start of the simulation for NorESM2 (second row) and as difference between NorESM2 and NorESM2fixed (third row). To cover differences of the initial climate state in NorESM2 and NorESM2fixed, the last row is calculated as double differences, meaning the difference of changes since the 1850s of the respective simulations.

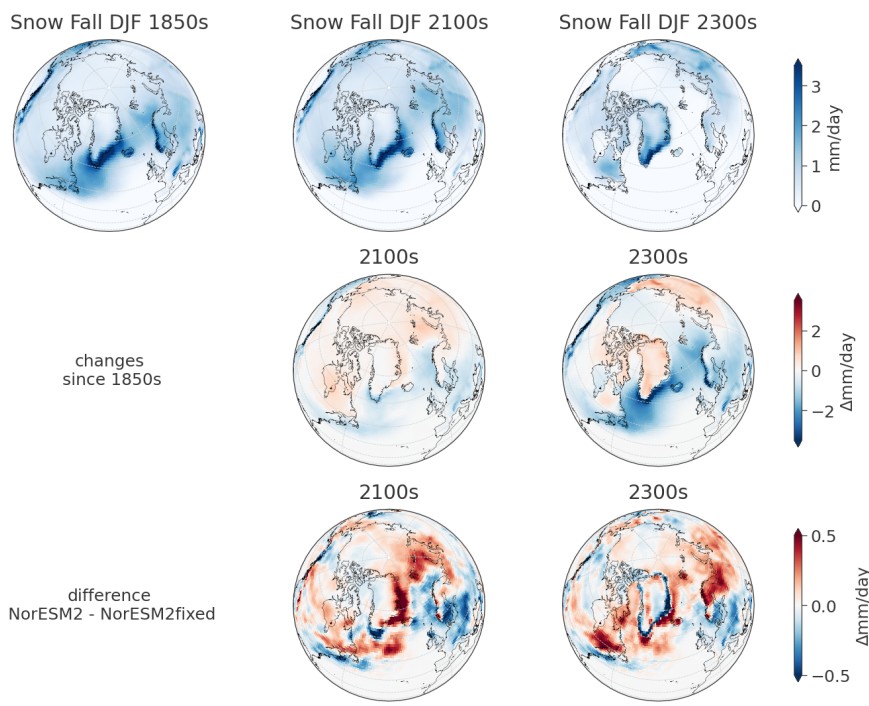

**Figure A6.** Arctic winter (DJF) snow fall with absolute values 1850, 2100 and 2300 for NorESM2 (top row), changes since the start of the simulation for NorESM2 (second row) and as difference between NorESM2 and NorESM2fixed (third row). To cover differences of the initial climate state in NorESM2 and NorESM2fixed, the last row is calculated as double differences, meaning the difference of changes since the 1850s of the respective simulations.

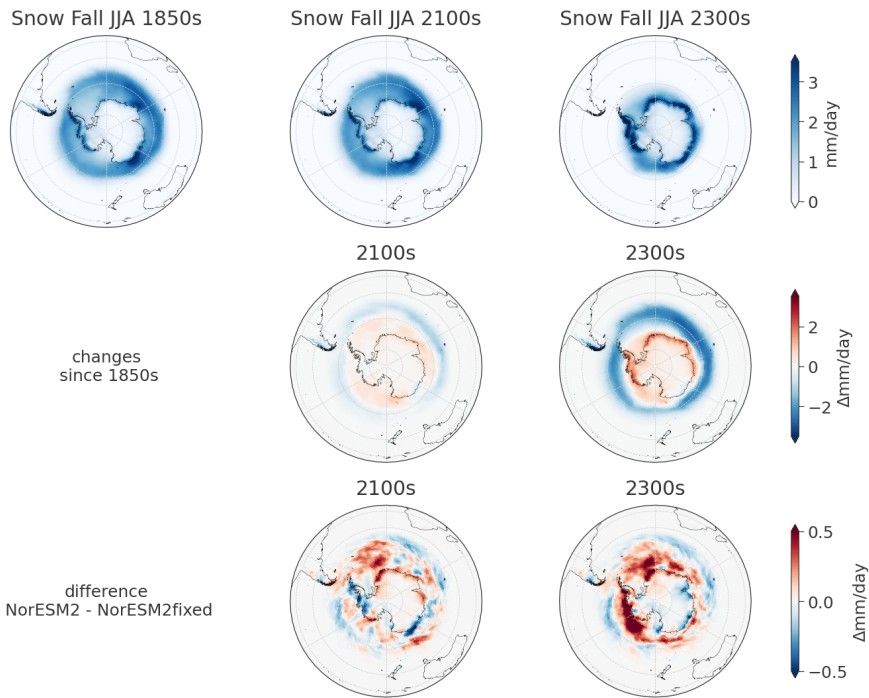

**Figure A7.** Antartic winter (JJA) snow fall with absolute values 1850, 2100 and 2300 for NorESM2 (top row), changes since the start of the simulation for NorESM2 (second row) and as difference between NorESM2 and NorESM2fixed (third row). To cover differences of the initial climate state in NorESM2 and NorESM2fixed, the last row is calculated as double differences, meaning the difference of changes since the 1850s of the respective simulations.

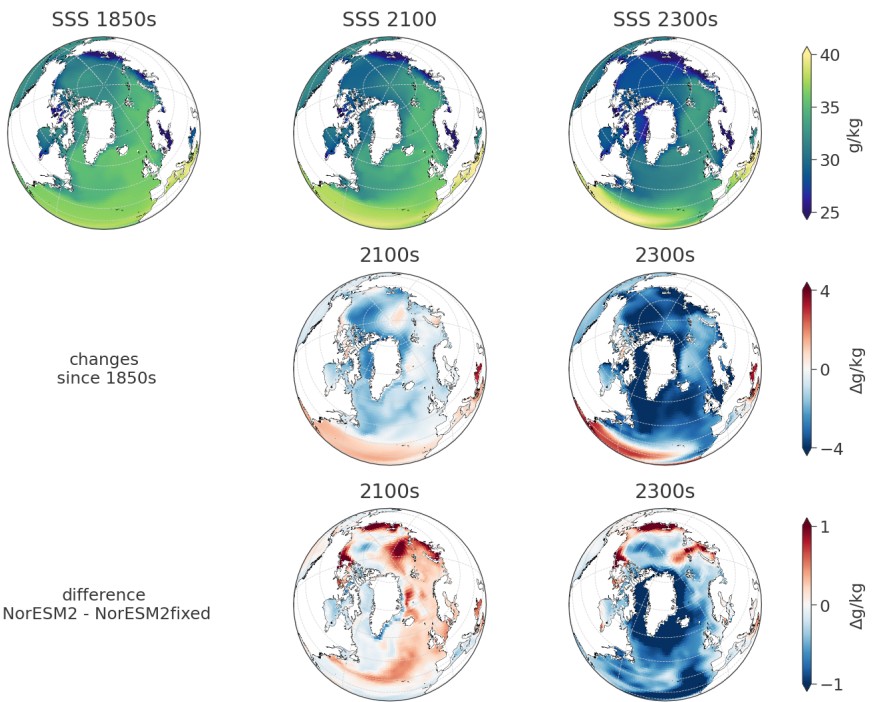

**Figure A8.** Sea salinity evolution as shown in Fig. 6 on Arctic projection. Absolute values 1850s, 2100s and 2300s for NorESM2 (top row), changes since the start of the simulation for NorESM2 (second row) and as difference between NorESM2 and NorESM2fixed (third row). To cover differences of the initial climate state in NorESM2 and NorESM2fixed, we show double differences here, meaning the difference of the anomaly in 2100s (and 2300s) for NorESM2 and NorESM2fixed.

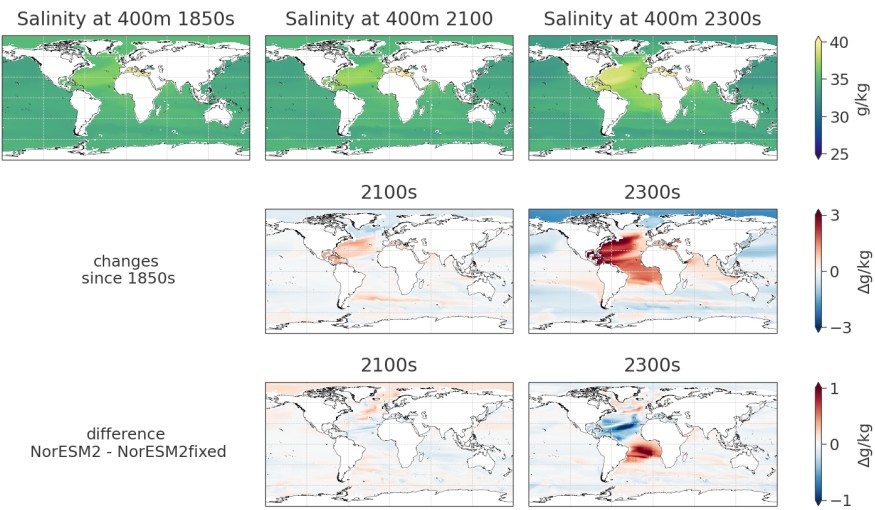

**Figure A9.** Sea salinity evolution as shown in Fig. 6 on 400 m depth. Absolute values 1850s, 2100s and 2300s for NorESM2 (top row), changes since the start of the simulation for NorESM2 (second row) and as difference between NorESM2 and NorESM2fixed (third row). To cover differences of the initial climate state in NorESM2 and NorESM2fixed, we show double differences here, meaning the difference of the anomaly in 2100s (and 2300s) for NorESM2 and NorESM2fixed.

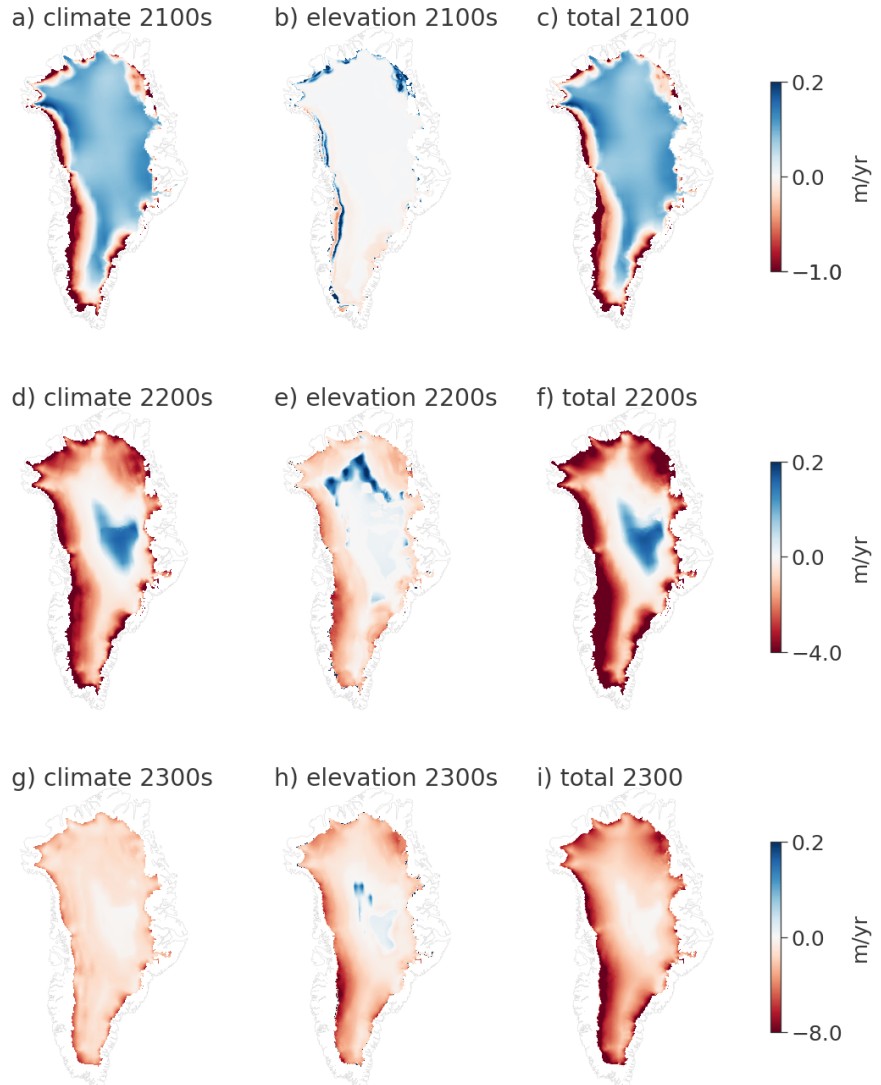

**Figure A10.** Surface mass balance anomalies compared to 1850s of the GrIS as shown in Fig.8 with different colour limits for each time interval. Surface mass balance anomaly in 2100 due to (a) climate, (b) elevation changes and (c) total anomaly. For 2200s and 2300s accordingly: with climate contribution to SMB anomaly in (d) 2200s and (g) 2300s. SMB changes since 1850s due to ice sheet elevation changes in (e) 2200s and (h) 2300s. Total SMB anomaly since 1850s in (f) 2200s and (i) 2300s. Gray outline marks the outline of Greenland.

*Author contributions.* AB acquired funding. HG set-up and performed the NorESM2 simulations. KH post-processed and analysed the simulation output with support of AB and HG. KH wrote the manuscript together with HG and AB.

*Competing interests.* The authors declare they have no competing interests.

*Acknowledgements.* We thank Michele Petrini, Miren Vizcaino, Ina Nagler, Aleksi Nummelin and KeyCLIM project participants for discussions and suggestions that supported the analysis of the simulations. High-performance computing and storage resources were provided by Sigma2 - the National Infrastructure for High Performance Computing and Data Storage in Norway through projects NN9560K, NN9252K, NN2345K, NN8006K, NS9560K, NS9252K NS2345K, NS9034K and NS8006K. HG has received funding from the Research Council of Norway under projects 324639, 270061, 352204 and 350390. Analysis and figures where done in an Python 3.9 environment with packages including numpy, xarray, netCDF4, matplotlib and cartopy. We thank the Norwegian Climate Centre for providing NorESM2 data for CMIP6 and the Earth System Grid Federation (ESGF) for archiving the CMIP data and providing access. This research has been supported by the Research Council of Norway under the KeyClim project (295046).

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
