# Peer review of "Limited global effect of climate-Greenland ice sheet coupling in NorESM2 under a high-emission scenario"

_EGUsphere, 2024_

## Referee Comment (RC1)

General Comments:

The authors present an extended high-emission scenario run with the coupled climate – ice sheet model NorESM2-CISM2.1. The study is the first to use NorESM with an interactive ice sheet model and adds nicely to other coupled climate – ice sheet and extended future warming scenario studies. Uncertainties in future projections of the Greenland ice sheet and its effects on the climate system are large. Therefore, it is desirable that more Earth system models include interactive ice sheet components. Multi-centennial simulations are necessary to account for the long time-scale processes in the land ice system.

The text is well-written and clearly structured. The figures are good. However, the text sometimes refers to fields (e.g. barotropic stream function or precipitation) for which no figures are included. It would be helpful to include such figures, maybe in the appendix. The text becomes a bit repetitive sometimes, especially in the discussion. The low climate sensitivity of the model and the corresponding small amount of land ice melting are pointed out too often. I think it is sufficient to do this in the introduction and then in the discussion again when necessary. With respect to the low melt rates, it may be useful to add some words about the relative cooling around Greenland compared to the uncoupled simulation (Fig2) and the potential implications of this relative cooling for the ice sheet melting in the discussion.

This study provides results with a newly coupled climate – ice sheet model. Substantial conclusions are made. The methods are clearly outlined and the results are sufficient to support the conclusions. Focusing on climate – ice sheet interactions, the manuscript should be suitable for publication in ESD, with minor revisions to be made.

Detailed Comments:

L11: "low weak amplification" – Do you mean "weak polar amplification"?

L39: "own" – better rephrase: "include".

L47-49: Ackermann et al., 2020 (GRL) did similar simulations with their Earth system model including an interactive Greenland ice sheet. You may consider to add them here.

L69: Please add a reference for the used configuration "NorESM2-MM".

L94f: Why is the orography updated every 5 years but surface types are updated annually? I understand that this publication focusses on the model results and that a detailed model description is done separately. However, I find it difficult to understand the workflow of the coupling procedure.

L104f: How is the horizontal and vertical spreading of freshwater done? Could you elaborate more on this or provide a reference? A second question: is there a heat flux associated with the solid and liquid runoff?

L117f: "stand alone" – should be: "standalone".

L122f: You may consider to move this sentence to the beginning of section 2.

Fig1d: You refer to the sea-level curve as "cumulative sea level contribution" but in the caption it says "Greenland ice mass changes." Does this curve show the ice loss or the actual sea level rise including other effects like thermal expansion? Please clarify.

Fig1h: It may be more intuitive to invert the y-axis to see the decline (less negative numbers) in line with the other timeseries.

L159: How is ice discharge treated differently in NorESM2fixed compared to the control simulation? I understand the different treatment between NorESM2 and NorESM2fixed but thought that the treatment in NorESM2 is the same as in the control simulation. Please clarify.

Fig2 third row: With "anomaly" you mean the changes since the 1850s of the respective simulation? Please clarify.

L166: It would be helpful to include a figure similar to Fig2 for the barotropic stream function.

L167: Please clarify, that you mean the barotropic stream functions in NorESM2 and NorESM2fixed that diverge from the control run and converge to each other.

L168: "… lower minimum values …" – better rephrase: "weaker gyre circulation".

L182: "Around the Greenlandic coastline" – better rephrase: "Around the coastline of Greenland".

L182-184: Where does the cooling around Greenland come from?

L185: Should probably be Fig3.

L196ff: This sentence seems a bit misplaced here in the results section. I suggest to move it to the introduction.

L200: "… is largely staying the same …" – better rephrase: "is largely unchanging".

L200ff: You may consider to add a figure for precipitation.

L207: "… during winter …" – which winter? Better use DJF / JJA etc.

L207: Like above: same for "summer precipitation".

Fig4: Does the figure show annual mean values? Please clarify.

L214: Do you annual mean sea surface temperatures? Please clarify.

L219f: "Between 2100s and 2300s increase by … since 1850s". Unclear formulation: Do you mean temperature increase between 2100s and 2300s or temperature increase since 1850s? Please clarify.

L221: Remove "still."

L224f: "The North Atlantic (above 50°N) and Arctic Ocean are becoming fresher" – To which simulation are you referring?

L226: " … are visible in a more saline …" – The differences are barely visible. You may think about adjusting the colorbar and adding a north polar stereographic projection, maybe as supplement.

L229: "… advancing divide …" – better rephrase: "increasing meridional gradient".

L230-232: You may consider to include a figure for salinity at depth.

L238f: "… covers more area …" very difficult to recognize in Figure 6. If I understand correctly, maximum sea ice extent in the Arctic by 2100 is the light blue solid line in the upper panels. By eye it looks to me as NorESMfixed has a larger sea ice area but differences are small anyway. Maybe drop this sentence or include a more distinct figure.

L242f: "… ice free during summer and nearly all-year ice free… (see Fig1) with a mean annual sea ice area of less than …". Otherwise the next sentence beginning with "The remaining winter sea ice …" sounds contradicting.

L246: What do you mean with changes of GrIS in NorESM2fixed? Greenland does not change in this simulation, does it?

L250: "… additional melt areas starts" – should be: "start".

L253: From Figure A1 it looks like SMB is always below 8 m/yr everywhere. Do you mean seasonal accumulation rates in the text? Please clarify.

L257: It would be helpful to include figures for ice velocity.

L260: "… direct SMB comparisons …" – should be: "comparison".

L260-262: "As explained in section 2, … This has been improved …" – consider to move this part to discussion.

L268f: Why "lack of SMB classes"? I thought, the classes are the same for both simulations.

L270-274: Refer to Figure A1.

L276f: ablation is less pronounced – To me it looks like the anomaly in 2200 is much more pronounced than in 2100. Please clarify.

L278: " there is less or decrease in accumulation…" – I find this sentence confusing. Should this mean "there is less of decrease", meaning that the rate of change becomes smaller? Please clarify.

L279f: Ablation increases compared to 1850.

L294: Consider to rephrase: "Arctic amplification factor is 0.2 higher in … than in …"

L301: Should be "in."

L307: "… and are hence harder to compare."

L307ff: Consider adding "by the year 2300."

L310: "lack of calving" – Do you mean an actual calving scheme with an iceberg model? In the method section you mention a calving parametrization for the ice sheet model.

L313: "Underestimating"

L313: What do you mean with "suggested" in this sentence?

L323f: "… not be developed enough… " What development do you mean here?

L343: Remove "and"

L353: For comparison it would be helpful to add the SAT increase in NorESM2 between 2015 and 2100.

L355: " … air temperature changes …" Remove "changes"

L356f: Repetition: you already mentioned several times the low CS of NorESM. Please consider to point this out only where necessary.

L359ff: Maybe the other way around: "Low GrIS mass changes allude to a low Arctic amplification and …"

L363: "This emphasizes …"

L363: "… of coupled climate – ice sheet models …"

L371: "becomes"

L373: remove "time"

L379: " – which contrasts model studies with EMICs"

---

## Author Comment (AC1)

**REVIEW#1**

This is the author's reply to RC1. We kept the general comments from the referee in original and added answers for each aspect directly below in blue.
The detailed comments will be addressed when preparing the revised manuscript.

General Comments:
The authors present an extended high-emission scenario run with the coupled climate – ice sheet model NorESM2-CISM2.1. The study is the first to use NorESM with an interactive ice sheet model and adds nicely to other coupled climate – ice sheet and extended future warming scenario studies. Uncertainties in future projections of the Greenland ice sheet and its effects on the climate system are large. Therefore, it is desirable that more Earth system models include interactive ice sheet components. Multi-centennial simulations are necessary to account for the long time-scale processes in the land ice system.
The text is well-written and clearly structured.
The figures are good.
Thank you for this summary and feedback!
We see the manuscript improving by including and addressing your comments.

However, the text sometimes refers to fields (e.g. barotropic stream function or precipitation) for which no figures are included.
It would be helpful to include such figures, maybe in the appendix.
L230-232: You may consider including a figure for salinity at depth.
L257: It would be helpful to include figures for ice velocity.
The barotropic stream function is shown in Figure 1h as a minimum of an area. We will include a reference to the subplot in the text and try to make it more clear that the first part of the results is explaining evolution of globally averaged values, and not about local, spatial distributed changes.
We are considering if additional figures for spatial precipitation patterns, salinity at depth and ice velocity add to the story and comprehension. We might add more figures.

The text becomes a bit repetitive sometimes, especially in the discussion. The low climate sensitivity of the model and the corresponding small amount of land ice melting are pointed out too often. I think it is sufficient to do this in the introduction and then in the discussion again when necessary.
We fully agree. We will go through the document and combine the mentions of low climate sensitivity where it is important and remove the other appearances.

With respect to the low melt rates, it may be useful to add some words about the relative cooling around Greenland compared to the uncoupled simulation (Fig2) and the potential implications of this relative cooling for the ice sheet melting in the discussion.
This is a very valid point. We will include some words about this in the manuscript.
The difference in surface air temperature stems here from the additional freshwater influx around the coast of Greenland. This increases ocean stratification and reduces vertical heat exchange leading to surface cooling.
This does not explain the initial lack of melting (which is due to the cold initial bias (discussed in Goelzer et al., (Disc.)).

This study provides results with a newly coupled climate – ice sheet model. Substantial conclusions are made. The methods are clearly outlined and the results are sufficient to support the conclusions. Focusing on climate – ice sheet interactions, the manuscript should be suitable for publication in ESD, with minor revisions to be made.

Thank you for this summary. We hope we can address your comments appropriately.

Detailed Comments:

We will include all minor comments and only address more detailed comments in the reply here.

L94f: Why is the orography updated every 5 years but surface types are updated annually? I understand that this publication focuses on the model results and that a detailed model description is done separately. However, I find it difficult to understand the workflow of the coupling procedure.

These are two separate processes in the model. The update of surface types happens in the land model CLM (at runtime) as soon as an update is available from the ice sheet model (yearly). Update of the surface topography and surface roughness for the atmosphere model CAM is an asynchronous process by modifying the restart files.

We will add a sentence here to make this point more clear.

L104f: How is the horizontal and vertical spreading of freshwater done? Could you elaborate more on this or provide a reference? A second question: is there a heat flux associated with the solid and liquid runoff?

Yes, the energy needed to melt ice is taken from the ocean heat reservoir.

We will describe the treatment of freshwater fluxes in more detail in the revised version, including the spreading function, conversion to salt fluxes and energy conservation.

L159: How is ice discharge treated differently in NorESM2fixed compared to the control simulation? I understand the different treatment between NorESM2 and NorESM2fixed but thought that the treatment in NorESM2 is the same as in the control simulation. Please clarify.

Exactly. NorESM2 and control have the same discharge treatment due to the additional ice sheet model coupling. NorESM2fixed is different. We will rephrase the sentence to avoid confusion.

L182-184: Where does the cooling around Greenland come from?

We addressed this in a reply to one of the general comments - see further up.

L310: "lack of calving" – Do you mean an actual calving scheme with an iceberg model? In the method section you mention a calving parametrization for the ice sheet model.

Yes, we mean a more physical or complex calving scheme that would e.g. respond to ocean warming, increased runoff or ice thinning. There is only a simple scheme applied to keep the ice margins within present day ice boundaries and to remove all floating ice.

We will revise this statement accordingly.

L353: For comparison it would be helpful to add the SAT increase in NorESM2 between 2015 and 2100.

True. We will add this value for NorESM2

Goelzer, H., Langebroek, P. M., Born, A., Hofer, S., Haubner, K., Petrini, M., Leguy, G., Lipscomb, W. H., and Thayer-Calder, K.: Interactive coupling of a Greenland ice sheet model in NorESM2, EGUsphere [preprint], https://doi.org/10.5194/egusphere-2024-3045, 2025.

---

## Author Comment (AC2)

**REVIEW#2**
This is the author's reply to RC2. We kept the general comments from the referee in original and added answers for each aspect directly below in blue.
The detailed comments will be addressed when preparing the revised manuscript.

In this paper, the authors analyze the first NorESM simulation including an interactive Greenland ice sheet model. The authors indicate a minor impact of interactive model on the global climate dynamics simulation. They do find differences in Arctic climate by 2100 and a large impact of the elevation-melt feedback on the overall ice sheet mass loss
Thank you for this summary and detailed feedback! We see the manuscript improving by including and addressing your comments.

**General comments**
The results are very interesting since there are very few examples of this type of coupled ice sheet-climate simulations. However, the analysis of results is difficult to follow as there are too many loose numbers in the text – these could be presented in tables instead – and with inconsistent metrics – sometimes ranges or extremes are presented, sometimes mean values with standard deviations -. The paper would benefit of some assessment of statistical significance of the differences between simulations (e.g, Figure 2), as well of (attempt of) explanations of these differences.
Thank you for this comment. We will go through the text and make the numbers more coherent. We only have three simulations and mainly compare time periods of 20yr means. We will try to include significance where it is possible and meaningful.

In the conclusions, the authors highlight how their results differ as those from previous work from EMICs. It would be interesting that the authors provide more detail on that.
EMICs differ from GCM in resolution and in complexity of parameterized processes. It is outside the scope of this study to analyze and discuss why the results are different. We can add information about the general differences in model complexities in the text.

The design of the simulations needs some more clarity, taking some parts of the model description (submitted) here where they are relevant to explain the results. For instance, Figure 2 does not provide information about initial NorESM2-NorESM2fixed (temperature, surface elevation) differences (for a pre-industrial or 1850 climate). Also, the treatment of meltwater fluxes in the "fixed" simulation is not clear; perhaps these fluxes could be compared in the manuscript.
We will consider adding information from the model description where it improves clarity. However, we would like to minimize repetition if possible (these variables are discussed in Goelzer et al., (Disc.))

**Title and main conclusion**
The title can be misleading, as it refers to "Limited global effect of climate-Greenland ice sheet coupling (…)". The conclusions rather refer to the limited effects of Greenland ice sheet change on the global climate. In the title the word climate is first, and the coupling could be uni-directional ("one-way"). Therefore, the title can read as if climate change does not affect Greenland or global sea levels when both climate and ice sheet are modelled together within NorESM … In general, I would highlight more the added value of the coupling

in title and conclusions, e.g., along the lines of mapping climate and (land, sea) ice change with a single model that permits to establish direct connections within the Earth System.

This is an interesting point. We are considering changing the title to avoid misleading information. At the same time it is important to not generalize our findings and state that there is never an effect of the added Greenland ice sheet component to the climate. It might just be NorESM or this particular setup.

**Figures**
Please add statistical significance. I suggest increasing the size, e.g., until at least page width. For Figure 2, it would be interesting to zoom in on the Greenland area/high latitude differences, and to relate with elevation differences.

We can unfortunately not increase the figure size due to the journal's style guidelines. Figure 7 tries to show the elevation and resulting SMB changes. We don't see the value of adding a zoom in SAT figure to the manuscript.

**Results**
Please introduce the structure of this long section. Also, consider adding numbered subsections. The differences in Arctic climate in the *fixed versus coupled simulations are a very interesting result, it would be great if the authors could go into more depth there.

This is a very good suggestion. We will structure this section better with more clear headers and will add a few more sentences to explain this.

Table 1 – please add standard deviations. Can you indicate which of the coupled versus fixed differences are statistically significant?

We will try an analysis of the table and see if there is significance, to put the numbers in bold and if there isn't any, to add a statement about this in the caption.

Line 146 Figure 1d seems to show a trend for the pre-industrial simulation? Can you quantify this and comment on it?

The trend is a desired effect of the initialisation: to reproduce an observed mass loss over the historical period. This is described in some detail in Goelzer et al., (Disc.). We will add a short description to clarify that.

Line 148 I don't see a graph or table in support of the global SST analysis
Lines 200-212: precipitation, is there a figure/table in support of "staying the same"? Also, numbers in the text can be replaced by a table.

This is similar what reviewer#1 added: we are considering adding a figure for precipitation. Figure 4 is already showing spatial comparisons of SST.

**Greenland ice sheet changes:**
There is not much analysis of surface mass balance or energy components here, is there any reason not to do this?

Thank you for this comment. There is already an evaluation of the SMB in the companion paper Goelzer et al., (Disc.) and we did not see an additional value to the story of our manuscript by including a detailed analysis of SMB and energy here. Further, this would only pertain to the coupled experiment and not to the comparison of NorESM and NorESMfixed, which is your focus in this paper.

Please increase size and add elevation contour lines to Figure 7, to illustrate changes in equilibrium line altitude. Perhaps interesting to add a figure/table with evolution of the mean equilibrium line altitude? (can be calculated from the ablation area % and the function of cumulative area below a certain elevation, or "hypsometric curve")

Given the fixed figure sizes in Earth System Dynamics, the only way to increase the size of individual panels is by changing their order. We prefer not to do that because the current setup follows a certain logic that we believe is important to keep.

The subplots show mainly SMB anomalies. Adding the ELA would likely add to confusion, because the ELA would be based on the absolut SMB, not the SMB differences shown in the Figure 7. Further, a detailed discussion of the SMB is not the scope of this manuscript: to compare differences between coupled and uncoupled experiments. However, a detailed SMB discussion that you would like to see can only be done for the coupled simulation. A more detailed analysis is done in Goelzer et a., (Disc). We can add a comment referring to that paper for the interested reader.

Figure 7h: there seems to be a strange "corner" along the equilibrium line altitude, do you know what causes this?

We cannot identify exactly which strange corner you refer to. In general, there are a few visible artefacts in some of the figures that are due to the downscaling and possibly the anomaly calculations.

We did not reply to minor comments here but will include and address them in the revised manuscript.

References:

Goelzer, H., Langebroek, P. M., Born, A., Hofer, S., Haubner, K., Petrini, M., Leguy, G., Lipscomb, W. H., and Thayer-Calder, K.: Interactive coupling of a Greenland ice sheet model in NorESM2, EGUsphere [preprint], https://doi.org/10.5194/egusphere-2024-3045, 2025.

---

## Referee Report (RR1)

Thank you for your reviewed manuscript. I see the authors did not incorporate much of my suggestions to reduce the abundance of numbers in the text, avoid detailed visual description of figures (the reader can do that by themself), add statistical significance (table, figures) and/or standard deviations together with (difference of) means (table, text), etc. In my view, those suggestions relate to common practice in scientific literature, please reconsider whether they are valuable to elevate the quality of the manuscript.

I have a couple of concrete requests:

- Figure with analysis of role of elevation change: I am surprised that the amount of SMB change due to climate change is much less in 2300 than 2200, can you please check?

- Analysis of velocity changes: can you please give proof of this, e.g., add a figure?

- The authors highlight the contribution of the elevation feedback to melt in the abstract. Can you please add a table with the integrated SMB with/without elevation feedback?

- Please check grammar and spelling e.g., through dedicated software.

---

## Author Response (AR3)

Dear editor,

We added a figure into the supplementary showing standard deviation of the surface air temperatures of the control run (pre-industrial forcing over 450 yrs). We further show the ratio of the absolute changes in air temperature of the coupled run (NorESM2) and the control standard deviation. The SAT changes by 2100s and 2300s are over 15 and 35 times the standard deviation of the control run representing internal model variability.

We added these sentences into the manuscript (line 179ff, also referring to the figure):
*Overall, the internal model variability is smaller than the changes due to climate forcing (Fig. A1). Comparing standard deviation of the climate change signal in NorESM2 by 2100s exceeds the 15 sigma range of inter-annual variability in some places. Changes by 2300s are locally over 30 times the inter-annual variability. This emphasizes that the changes seen in the simulation over time are outside of the internal model variability.*

We would like to add that variability for globally averaged variables (like SAT, SST, SSS etc.) can be seen in Figure 1 - where both control and NorESM2 results are shown.

We hope we could address your suggestion with these adjustments.

Kind regards,
Konstanze Haubner
Heiko Goelzer
Andreas Born

[Figure]

**Figure A1.** Left: SAT standard deviation (SD) of the control run (pre-industrial forcing from 1850 to 2300). Middle and right show the ratio of the absolute SAT changes (by 2100s and 2300s respectively) and the SD of the control run.